# ERεLELA: Exploration in Reinforcement Learning via Emergent Language Abstractions

## Abstract

The ability of AI agents to follow natural language (NL) instructions is important for Human-AI collaboration. Training Embodied AI agents for instruction-following can be done with Reinforcement Learning (RL), yet it poses many challenges. Among these is the exploitation versus exploration trade-off in RL. Previous works have shown that using NL captions as state abstractions can help address this challenge. However, NLs descriptions have limitations in that they are not always readily available and are expensive to collect. In order to address these limitations, we propose to use the Emergent Communication paradigm, where artificial agents learn an emergent language (EL) in an unsupervised fashion, via referential games. Thus, ELs constitute cheap and readily-available state abstractions. In this paper, we investigate (i) how EL-based state abstractions compare to NL-based ones for RL in hard-exploration, procedurally-generated environments, and (ii) how properties of the referential games used to learn ELs impact the quality of the RL exploration and learning. We provide insights about the kind of state abstractions performed by NLs and ELs over RL state spaces, using our proposed Compactness Ambiguity Metric. Our results indicate that our proposed EL-guided agent, entitled ERεLELA, achieves similar performance as its NL-based counterparts without its limitations, and is competitive with state-of-the-art approaches in hard-exploration RL. Our work shows that RL agents can leverage unsupervised EL abstractions to greatly improve their exploration skills in sparse reward settings, thus opening new research avenues between Embodied AI and Emergent Communication.

## 1 Introduction

Natural Languages (NLs) have some properties, such as compositionality and recursive syntax, that allow us to talk about infinite meanings while only using a finite number of words (or even letters, or phonemes, etc.). In other words, it enables us to be as expressive as one might need. However, it may be interesting sometimes to use language to abstract away from the details and only focus on the essence of a specific experience, or a specific sensory stimulus. Thus, even though NLs can sometimes be used with high expressiveness, they can also work as abstractions. This can be observed when using the same or similar referring expressions to talk about superficially distinct, but causally- or semantically-related situations. This is possible because (natural) language abstractions have been shaped through (natural/human) communication processes to capture such relationships.

Tam et al. (2022) investigated leveraging such abstractions for training Reinforcement Learning (RL) agents in simulated 3D environments. In effect, some unique NL utterances can be found to refer to a lot of semantically-similar but visually-different observations of the agent. For instance, the utterance 'one can see a purple key and a green ball' can refer to many first-person perspectives of an embodied agent, irrespective of some orientational and positional aspects of that embodied agent. Tam et al. (2022) referred to that phenomena as compacting/clustering a state/observation space. Indeed, NL abstractions are in effect segmenting the state space into a set of less-detailed but more-meaningful sub-spaces. We employ the term meaningful here with respect to the task that the embodied agent is possibly trained for. For instance, if the task consists of picking and placing objects, then it is meaningful for utterances to contain information about objects and places, but not

so much to contain information about other agents in the environment, if any. Tam et al. (2022) and Mu et al. (2022) provided some arguments towards the compacting/clustering assumption of NLs. In their experiments, they employed NL oracles to build abstractions over 3D and 2D environments. They consist of NL captioner systems that take as inputs a complex state/observations, as well as some private underlying information from the RL environment, to return an NL caption highlighting the main features visible in the input observation. This is necessary in order to experiment with NLs in a controlled and easy manner, as it skirts the expensive problem of collecting NLs utterances from human participants. Those NL-based abstractions were then leveraged in state-of-the-art exploration algorithms, such as Random Network Distillation (RND - Burda et al. (2018)) and Never-Give-Up (NGU - Badia et al. (2019)), which can be difficult to deploy compared to, for instance, a count-based method. Indeed, count-based methods involves (i) fewer moving parts (.e.g state-count buffer versus e.g. RND's random and predictor networks, and predictor optimizer), (ii) they can be deemed simpler to implement (no tricks required on the contrary to RND's tricks like reward normalization and observation clipping and normalization that are critical), and (iii) they involve fewer hyperparameters to finetune (e.g. only a reward-mixing coefficient on the contrary to e.g. RND's reward mixing coefficient, architectures of random and predictor networks, hyperparameters of the predictor optimizer, and different intrinsic and extrinsic discount factors).

Thus, this work aims to simplify the process of using languages as abstractions and to address the limitation of using NLs, which are expensive to harvest and not necessarily the most meaningful abstractions for any given task. Indeed, instead of state-of-the-art exploration algorithms, we show that simpler count-based approaches combined with language abstractions can be leveraged for hard exploration tasks. And, to remove the reliance on NLs, we look at the field of Emergent Communication (EC) (Lazaridou & Baroni, 2020; Brandizzi, 2023) which have shown that artificial languages, referred to as Emergent Language (EL), can emerge through unsupervised learning algorithms, such as Referential Games (RGs) and variants (Denamganaï & Walker, 2020a), with structure and properties similar to NLs (Brandizzi, 2023; Rita et al., 2020).

Thus, we propose **Exploration in Reinforcement Learning via Emergent Language Abstractions (EReLELA)**, an exploration method that leverages the efficiency of an Emergent Language (EL) to generate a discrete, intra-life intrinsic reward signal. We demonstrate EReLELA's effectiveness across hard exploration environments, achieving **new state-of-the-art performance** in the 3D MiniWorld FullMazeS5 environment, and outperforming recent strong methods like ETD (Jiang et al., 2025) and DEIR (Wan et al., 2023). Crucially, our core contribution shifts the focus from merely utilizing language to controlling its abstractive quality: we show the Relative Expressivity (RExpr) of the EL is the critical determinant of RL exploration success ($R^2 \approx 0.52$). Furthermore, we introduce the Compactness Ambiguity Metric (CAM) to measure the quality of the abstractions performed by languages. It relies on evaluating the state compacting/clustering qualities of languages and is computed on video-like stream of frames and their captions in the evaluated language. To the best of our knowledge, this is the first metric of this kind. Using the CAM, we provide the first analysis of the structure of these effective ELs abstractions, revealing they consistently converge on a **minimal, task-relevant abstraction** (a Shape-only bias) rather than a compositional natural-language-like one. This work establishes the value of explicitly controlling the fidelity and analyzing the nature of ELs abstractions for robust and efficient RL exploration.

## 2 BACKGROUND

### 2.1 EXPLORATION VS EXPLOITATION IN REINFORCEMENT LEARNING

An RL agent interacts with an environment in order to learn a mapping from states to actions that maximises its reward signal. Initially, both the reward signal and the dynamics of the environment (the impact that the agent actions may have on the environment) are unknown to the agent. It must explore the environment and gather information. Yet, all the while it is exploring, it cannot exploit the best strategy that it has found so far to maximise the known parts of the reward signal. This dilemma is known as the Exploration-vs-Exploitation trade-off of RL (Sutton & Barto, 2018; Kaelbling et al., 1996). This dilemma is not the only challenge, as it can even get worse, especially in sparse reward environments where the reward signal is mainly zero most of the time. This context makes it very difficult for agents to learn anything, because RL algorithms derive feedback (i.e. gradients to update their parameters) from the reward signal that they observe from the environment. It is referred to as

extrinsic reward signal because it comes from the environment. As the extrinsic reward is mostly zero in spare reward environments, agents must exploit another signal to derive information about the currently-unknown environment. This other signal can be found in relation to the observation/state space, as agents can learn to seek novelty or surprise around the observation/state space and attempt to manipulate it efficiently by choosing relevant actions. Focusing on this novelty, agents can harvest another feedback signal, that is referred to as intrinsic reward signal. Note that this intrinsic reward signal is very different from the extrinsic one, because it does not inform agents about the task they must perform in the environment. Ideally, though, it provides a dense signal they can use to start learning something about the environment and its dynamics. This is inspired by intrinsic motivation in psychology (Oudeyer & Kaplan, 2008). Exploration driven by curiosity/novelty might be an important way for children to grow and learn. Here, we focus on novelty to derive the intrinsic rewards but it could be correlated with e.g. impact (Raileanu & Rocktäschel, 2019), surprise (Burda et al., 2018) or familiarity of the state. The intrinsic reward signal is only a proxy for agents to start to make progress into learning about the environment and eventually, hopefully encounter some non-null extrinsic reward signal along the way. We refer readers to Appendix C for further details.

## 2.2 EMERGENT COMMUNICATION

Emergent Communication is at the interface of language grounding and language emergence. While language emergence raises the question of how to make artificial languages emerge, possibly with similar properties to NLs, such as compositionality (Baroni, 2019; Guo et al., 2019; Li & Bowling, 2019; Ren et al., 2020), language grounding is concerned with the ability to ground the meaning of (natural) language utterances into some sensory processes, e.g. the visual modality. While the compositionality of ELs has been shown to further the learnability of said languages (Kirby, 2002; Smith et al., 2003; Brighton, 2002; Li & Bowling, 2019), ELs are far from being 'natural-like' protolanguages (Kottur et al., 2017; Chaabouni et al., 2019a;b), and the questions of how to constrain them to a specific semantic or a specific syntax remain open problems.

The backbone of the field rests on games like the *Signalling Game* or *Referential Game (RG)* by Lewis (1969), where a speaker agent is asked to send a message to the listener agent, based on the *state/stimulus* of the world that it observed. The listener agent then acts upon the observation of the message by choosing one of the *actions* available to it, with the aim to perform the 'best' *action* given the observed *state*, where the notion of 'best' *action* is being defined by the interests common to both players. We refer readers to Appendix C.3 for more background about the stakes in Emergent Communication and relation to learning generalisable and possibly-disentangled representations (Xu et al., 2022; Denamganaï et al., 2023). Thus, this paper aims to investigate visual discriminative RGs as auxiliary tasks for RL agents.

We proceed with formally describing (visual) discriminative RGs. They have the listener action being to identify a target stimulus from a set of candidate stimuli, based solely on the speaker's message describing said target stimulus. Formally, let $\mathcal{D}_{RG} \subset \mathcal{S}$ be a dataset of stimuli and $\Sigma^\star$ be the set of all language utterance using vocabulary $\Sigma$. The discriminative RG involves two agents, a speaker $Sp : \mathcal{S} \mapsto \Sigma^\star$ that maps stimuli to emergent language utterances, and a listener $Li : \mathcal{S} \times \Sigma^\star \mapsto \mathbb{R}$ that evaluates compatibility between stimuli and utterances. The optimal speaker and listener functions are found solving:

$$Sp^\star, Li^\star = \underset{Sp, Li}{\operatorname{argmin}} J_{RG}(Sp, Li) \text{ s.t. : } J_{RG}(Sp, Li) = \mathbb{E}_{\mathcal{D}_{RG}^{K+1}}[\mathcal{L}(Li, Sp)] \quad (1)$$

with $\mathcal{L}(Li, Sp) = ((Li(s_i, Sp(s_0)))_{i \in [0,K]})$, and stimuli $(s_i)_{i \in [0,K]} \sim \mathcal{D}_{RG}^{K+1}$. Finally, $\mathcal{L}$ refers to a loss function, such as the Hinge loss more commonly :

$$\mathcal{L}((Li(s_i, Sp(s_0)))_{i \in [0,K]}) = max(0, \gamma - Li(s_0, Sp(s_0)) + \underset{i \in [1,K]}{max} Li(s_i, Sp(s_0))) \quad (2)$$

with target stimulus $s_0$, distractor stimuli $(s_1, \dots, s_K)$, and margin hyperparameter $\gamma$ (typically 1). This optimization objective ensures that the speaker generates discriminative utterances in the emergent language, and it prompts the listener to correctly associate target stimuli $s_0$ with their corresponding utterances $Sp(s_0)$.

## 3 METHODS

We start by presenting the EReLELA architecture that leverages EL abstractions in an *intra-life* count-based exploration scheme (cf. Appendix C) for RL agents, in Section 3.1. Then, acknowledging a gap in evaluating the state abstractions that different languages perform over different state/observation spaces, we introduce our Compactness Ambiguity Metric (CAM) that attempts to fill in that gap, in Section 3.2.

### 3.1 EReLELA ARCHITECTURE

The Exploration in Reinforcement Learning via Emergent Language Abstractions (EReLELA) architecture is a wrapper around any off-/on-policy RL algorithm (left in Fig. 1). It augments the standard reward signal by linearly combining the original extrinsic reward with an intrinsic reward derived from an EL. The core mechanism relies on an *intra-life*/episodic count-based exploration method (top, left, green block in Fig. 1).

Intuitively, EReLELA's RG speaker agent implements a hashing-like function (cf. Appendix C.2) that turns continuous and high-dimensional observations (such as pixel-based frames) into discrete, variable-length sequences of tokens. The intrinsic reward generator

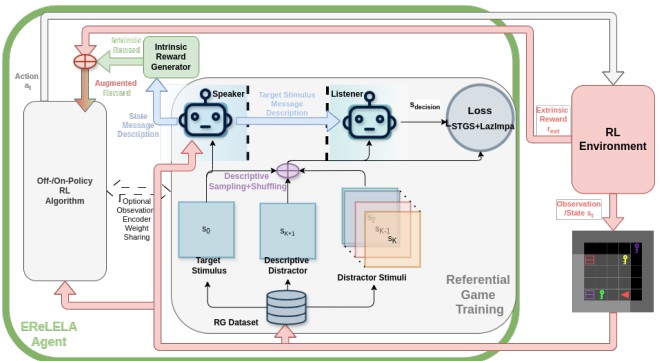

Figure 1: EReLELA agent in the context of the common RL feedback loop, detailing how the intrinsic reward generator leverages the state abstraction performed by the RG speaker agent to compute an intrinsic reward which is then linearly combined with the RL environment's extrinsic reward. The intrinsic reward generator consists of an intra-life count-based exploration method. In its most general form, EReLELA is a wrapper around any off-/on-policy RL algorithm. Optionally, the weights between the RL algorithm's observation encoder and the RG players' stimulus encoder may be shared, following an unsupervised auxiliary task framing (Jaderberg et al., 2016).

then counts the occurrence of these captions within the current episode to estimate state novelty, encouraging the agent to visit states that elicit novel linguistic descriptions.

EReLELA involves an unsupervised auxiliary task in the form of a (discriminative, here, or generative) RG to train in an unsupervised fashion the speaker and listener agents (center of Fig. 1), following the UNREAL architecture from Jaderberg et al. (2016).

Formally, we study a single agent in a Markov Decision Process (MDP) defined by the tuple $(\mathcal{S}, \mathcal{A}, T, \mathcal{R}, \gamma)$, referring to, respectively, the set of states, the set of actions, the transition function $T : \mathcal{S} \times \mathcal{A} \to P(\mathcal{S})$ which pro-

$$\max_{\pi} \mathbb{E}_{\tau \sim \pi} \left[ \sum_{t=0}^{T} \gamma^t \mathcal{R}(s_t, a_t, s_{t+1}) \right] \quad (3)$$

vides the probability distribution of the next state given a current state and action, the reward function $\mathcal{R} : \mathcal{S} \times \mathcal{A} \to r$, and the discount factor $\gamma \in [0, 1]$. The agent is modelled with a stochastic policy $\pi : \mathcal{S} \to P(\mathcal{A})$ from which actions are sampled at every time step of an episode of finite time horizon $T$. Following Eq. 3, the agent's goal is to learn a policy that maximizes the expected return over trajectories $\tau = (s_0, a_0, s_1, a_1, \ldots, s_T)$ generated by following policy $\pi$.

**Intrinsic Motivation.** We further define $\mathcal{R} = \lambda_{\text{ext}} \mathcal{R}^{\text{ext}} + \lambda_{\text{int}} \mathcal{R}^{\text{int}}$ as the weighted sum of the extrinsic and intrinsic reward functions, respectively, $\mathcal{R}^{\text{ext}}, \mathcal{R}^{\text{int}}$, with weights $\lambda_{\text{ext}}, \lambda_{int}$. Indeed, while the extrinsic reward is provided by the environment, the intrinsic reward is computed by

$$\forall t, \mathcal{R}^{\text{int}}(s_t | \tau_{<t}, \text{Sp}_{RG}) =$$
$$\begin{cases} 1 & \text{if } \text{Sp}_{RG}(s_t) \notin \text{Sp}_{RG}(\tau_{<t}) \\ 0 & \text{otherwise} \end{cases} \quad (4)$$

the *Intrinsic Reward Generator* (cf. Figure 1) using the output of the RG speaker agent. Formally, we define the RG speaker agent as the function $\text{Sp}_{RG} : \mathcal{S} \mapsto V^L$ where $V$ is the vocabulary and $L$ the maximum sentence length of the RG. Thus, as an intra-life count-based method, the EReLELA's intrinsic reward function takes as input the current state $s_t$ and is conditioned on all the previously-

observed states so far in the episode (as opposed to over the whole training process, referred to as *across-training* - cf. Appendix C), $\tau_{<t} = (s_k)_{k \in [0,t-1]}$, as shown in Eq. 4.

**Referential Game Training.** The validity of this count-based signal depends entirely on the quality of the abstractions performed by the RG speaker. Following the nomenclature proposed in Denamganaï & Walker (2020b), we employ a *descriptive object-centric (partially-observable) 2-players/$L = 10$-signal/$N = 0$-round/$K$-distractor* RG variant (cf. Figure 16 in Appendix H). The descriptiveness implies that the target stimulus is not always passed to the listener agent, but instead sometimes replaced with a descriptive distractor (cf. Appendix H for implementation details). The object-centrism is achieved via application of data augmentation schemes before feeding stimuli to any RG agent, following Dessì et al. (2021) but using Gaussian Blur transformation alone, as it was found sufficient in practice.

To ensure that our approach works across different language structures, we optimize the RG agents either of two distinct loss functions, both adapted for a differentiable Straight-Through Gumbel Softmax (STGS) communication channel, either the Impatient-Only STGS loss or the STGS-LazImpa loss, inspired from Rita et al. (2020) and detailed in Appendix H.1. The LazImpa-based loss prompts the RG agents to learn ELs that bear the property of Zipf's law of Abbreviation (ZLA - Zipf (2016)). ZLA is an empirical law found in most NLs which states that the more frequent a word is, the shorter it tends to be. Alternatively, we employ as comparison point the Impatient-only loss Rita et al. (2020) designed to prompt the listener agent to guess the target stimulus as early as possible when attending to the speaker's utterance, as opposed to solely guessing upon reading the EoS token at the end of the speaker's utterance.

**Training Protocol.** RG agents are trained periodically, every $T_{RG}$ gathered RL observations, on a buffer $\mathcal{D}_{RG}$ of recent observations to ensure the EL evolves alongside the RL agent's exploration frontier. Training is performed over a maximum of $N_{\text{RG-epoch}}$ or until a terminating condition is met, such as exceeding a validation RG accuracy threshold (more on this below). Optionally, the weights between the RL algorithm's observation encoder and the RG players' stimulus encoder may be shared, following an unsupervised auxiliary task framing (Jaderberg et al., 2016). We refer to the architecture with and without shared weights, respectively, as *shared* and *agnostic*.

**Controlling Exploration via Relative Expressivity.** A critical question in this setup is determining which properties of the RG training enable effective exploration. While the number of distractors $K$ during RG training appears important—with preliminary experiments suggesting higher $K$ and higher accuracy threshold generally aid performance (see Appendix E.2 and E.1 for details, respectively)—our analysis reveals that these parameters are poor predictors of success. Specifically, we find that the coefficient of determination between the RL success rate and the number of training distractors $K$ is low ($R^2 = 0.1915$).

Instead, we identify the Relative Expressivity (RExpr) of the EL as a more determining factor for exploration quality and, therefore, RL success. Relative Expressivity is the ratio, expressed as a percentage, between the number of unique sentences in the EL over the size of the buffer $\mathcal{D}_{RG}$. We call it relative because we cannot measure the true/asbsolute expressivity of the EL given the fact that we cannot know the number of unique stimuli in the buffer. Our results indicate a stronger correlation between the RExpr of the EL and the RL agent's coverage over the environment ($R^2 = 0.46$ - See App. F.4) and its success rate ($R^2 = 0.52$). This evidence suggests that simply increasing the number of distractors is insufficient; rather, controlling the Relative Expressivity is critical. An EL that is too coarse (low (relative) expressivity) fails to distinguish relevant state changes, collapsing the intrinsic reward signal. Conversely, an EL that is too specific may fail to group semantically similar states, thus inflating the intrinsic reward signal unnecessarily. EReLELA therefore relies on RG training protocols specifically tuned to further RExpr above a parameterised threshold (empirically found to be $> 40\%$) to ensure that the linguistic abstractions effectively guide the RL agent. More specific details about the RG and its agents' architectures can be found in Appendices G and H and our open-source implementation[1].

---

[1]HIDDEN_FOR_REVIEW_PURPOSE

## 3.2 Compactness Ambiguity Metric

The interplay between Relative Expressivity and exploration performance (Coverage) raises a fundamental question: what specific abstractions are these languages performing ? Standard metrics in Emergent Communication have not been addressing this question so far because temporally-correlated observations in dynamic environment have not been considered. To address this, we introduce the Compactness Ambiguity Metric (CAM).

**Intuition.** Let us consider an embodied agent navigating in an environment towards fulfilling a given goal. For instance, the goal could be to pick up a specific object from one of the rooms of a house filled with many objects of different shapes and colours. Let us consider the captions that myopic and astigmatic individual would produce when observing the agent's first-person viewpoint. Their captioning would only detail the colour of the closest visible object, failing to describe its shape due to astigmatism, and failing to detail anything about further away. This captioning is an example of state abstraction in this environment. Let us now consider the captions that a colour-blind and myopic individual would produce. Because of their colour-blindness, they would only describe the shape of objects, and restrict themselves to the closest object due to being myopic.

We now focus on the differences in captioning that they would produce when prompted with the very same embodied agent trajectory. Since those captionings are state abstractions, they must be ambiguous in the sense that each caption would refer to many states/observations. We would expect all those states that map to the same caption from either captioner to be temporally correlated to each other, at least, since the embodied agent does not teleport from one room to another, but rather moves step by step and its surroundings and observations maintain some consistency from one step to the next. In effect, captionings would be grouping/compacting together states that are temporally-correlated. Those groupings would be especially salient features when considering the captions over consecutive timesteps in the embodied agent's trajectory. For instance, all while the embodied agent is passing by and facing multiple *blue* objects, e.g. a ball and then a key, then we would expect the myopic-and-astigmatic captions to remain constant over many timesteps saying *'I can see a blue object'*. On the other hand, the colour-blind-and-myopic captions would group together states differently depending on which of the blue object is the closest at any given time, being constant firstly with *'I can see a ball'*, before then switching to *'I can se a key'*. Thus, we derive the intuition that state abstractions must be characterizable by the kind of compacting of states that they perform, and more precisely in terms of temporal correlations, i.e. for how many consecutive timesteps does a given caption remains unchanged.

As such, we propose the Compactness Ambiguity Metric (CAM) to measures the qualities of the state abstraction performed by languages. It relies on evaluating their compacting/clustering qualities over stimuli. It assumes temporally-correlated stimuli as inputs. For instance, inputs can be a set of video-like stream of frames and their captions. The CAM evaluates the language used in the captions. To do so, it sorts into different bins of an histogram the different captions. This sorting is based on the length of the time interval that each caption occupies over the video stimuli. For instance, the caption from time step $t$ to $t+k$ of a video may all be the same, over $k$ consecutive frames. Therefore it would be sorted into the histogram's bin corresponding to length $k$. This time interval length corresponds to a measure of the ambiguity of said caption. The longer the time interval is, the more (temporally) ambiguous the caption is. The metric assumes that the more ambiguous a caption is the more details it abstracts. We will discuss below how this assumption is imperfect, but still useful. Different time interval lengths will correspond to different qualities of abstractions. Thus, the resulting histogram yields a distribution of the qualities of the abstractions. Different languages create distinct abstraction histograms when computed over the same video stimuli. We can then compare these histograms by computing distance metrics. This allows us to quantify how different languages abstract things.

**Formalism.** A CAM measure consists of a distribution, represented by an histogram of $N$ bins, where $N$ is one of the two hyperparameters of the metric. We refer to the counts in the bins as CAM scores. The CAM takes as inputs (i) a video-like input framed as a dataset of $N_{\mathcal{D}}$ RL trajectories of length $T$: $\mathcal{D} = \{s_t \in \mathcal{S} | t \in [1, T \cdot N_{\mathcal{D}}]\}$, and (ii) a speaker agent whose utterances are in the language $l$ that we want to evaluate with the metric. We first define a language $l$ as a subset of $V^L$ where $V$ is a vocabulary with $|V|$ tokens and $L$ is the maximum length of each utterance/caption. Thus, for each language $l \subseteq V^L$, we define a speaker $\mathrm{Sp}_l : \mathcal{S} \mapsto V^L$, such that $\mathrm{Sp}_l(\mathcal{D}) = l$. We refer the reader to Algorithm 1 and Appendix D for details on the CAM's computation. We show in Appendix F.1 that this metric hast internal validity, meaning that (i) CAM enables us to discriminate between different

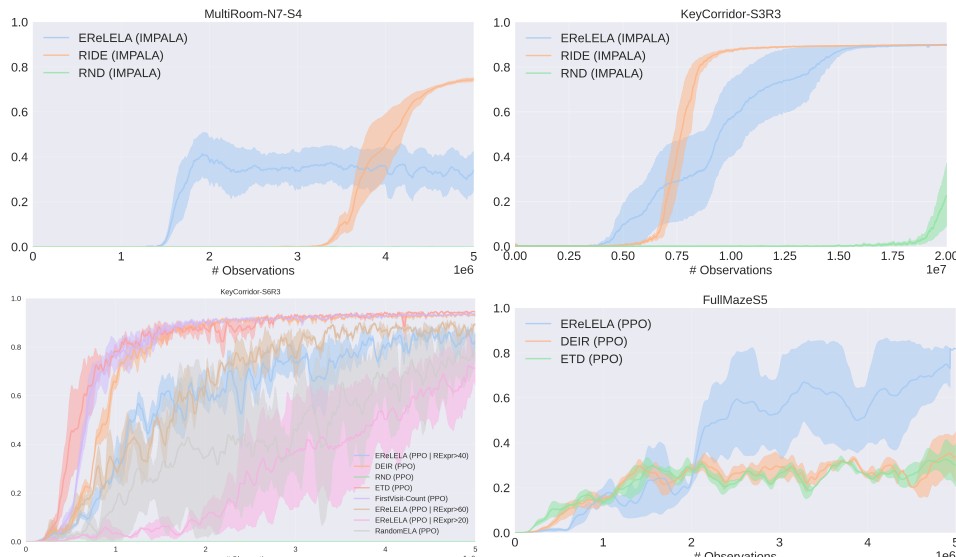

Figure 2: Comparative RL performance (mean episodic return) across environments (MultiRoom-N7S4, KeyCorridorS3R3, KeyCorridorS6R3, FullMazeS5).

languages that are known to build different state-abstractions (e.g. synthetic languages that refers to all or only one specific attribute of objects, such as color or shape, used to caption a video stream that is an egocentric viewpoint of an agent randomly walking in a 3D room with many randomly-placed objects), and (ii) CAM maps languages without consistent state-abstractions (e.g. shuffled captions over a video stream) close to a null distribution histogram.

**CAM Distances.** As the CAM returns a distribution in the form of an $N$-binned histogram, many different distance metric could be computed between two such distributions. In this paper, we choose to define the CAM distance as an euclidean distance in $\mathbb{R}^N$ by considering the $N$ CAM scores (the count in each bin of the histogram) as vectors in $\mathbb{R}^N$.

## 4 EXPERIMENTS & RESULTS

This section details our experimental results across $2D$ symbolic and $3D$ pixel-based environments, respectively, from MiniGrid and MiniWorld (Chevalier-Boisvert et al., 2023). We structure our findings around 3 research questions, focusing on EReLELA's comparative performance and, critically, the role of EL abstractions and their structure, and the critical components of our method.

**RQ1: Comparison to state-of-the-art on hard exploration environments?**   Our main performance results across four challenging environments are summarized in Figure 2. We compare EReLELA against a set of strong PPO- (Schulman et al., 2017) or IMPALA-based (Espeholt et al., 2018) state-of-the-art algorithms and use the same base algorithm for fair comparison: **ETD** (Episodic Novelty Through Temporal Distance -  Jiang et al. (2025)) which uses temporal distance as a robust metric for state similarity and intrinsic reward computation, yielding current best results on both environments ; **DEIR** (Discriminative-model-based Episodic Intrinsic Reward -  Wan et al. (2023)) which **FirstVisit-Count**, which is a strong episodic, symbolic count-based method adapted from Henaff et al. (2023) and  Wang et al. (2023), serving as a an upper-bound first-visit-count-based comparison; **RIDE** (Rewarding Impact-Driven Exploration -  Raileanu & Rocktäschel (2019)) which encourages the RL agent to take actions that lead to significant changes in its learned state representation, proposing a comparison to a differently-principled approach ; and **RND** (Random Network Distillation -  Burda et al. (2018)) which computes intrinsic reward from the prediction error over continuous embeddings of observations via a randomly-initialised neural network. Results show that EReLELA is competitive with or exceeds the state-of-the-art across these benchmarks, irrespective of the base algorithm used.  Notably, EReLELA demonstrates a strong performance advantage in

the FullMazeS5 (MiniWorld $3D$) environment, achieving the highest mean episodic return and sample-efficiency. This result highlights its robustness to high-dimensional, visual observations, where traditional count-based methods would struggle due to most pixel-based observations being counted as unique and therefore wrongly inflating the intrinsic reward signal.

We remark that our proposed architecture's main differentiation feature from other approaches is that the state abstractions that EReLELA performs are discrete and periodically updated to efficiently **discriminate between the set of states recently encountered, as a whole**. This is thanks to the periodic RG training, performed on the latest states encountered by the RL agent, that optimises the EL to improve the ability of the speaker and listener agents to coordinate in efficiently discriminating between target state and (many) distractor states. This is different from, e.g. RIDE, RND, and ETD whose state abstractions are continuous and only periodically updated with respect to **each individual state's impact and/or novelty**. Neither update each state abstractions in relation to other states with more sublety than just normalisation, whereas EReLELA does. To some extent, DEIR can be argued to perform this periodic update as a whole as well, but it relies on continuous abstractions, whereas EReLELA relies on discrete ones.

**RQ2: What kind of abstractions do the emergent languages perform?** To answer this, we apply the Compactness Ambiguity Metric (CAM) over the MiniGrid environment and compute CAM distances against synthetic language baselines: a compositional Natural Language (NL) baseline (compositional around shape and color), a Color-only language, and a Shape-only language. We use a barycentric plot in Figure 3 to visualize the relative distances of the ELs' CAM score to the language baselines. The vertices of the triangle correspond to pure convergence toward that abstraction type. The barycentric plots show the evolution of the barycenter over the course of RG training epochs (in parallel of RL steps). Results evidence that the ELs abstractions are consistently distinct from the Compositional NL baseline. Instead, the ELs strongly latch onto a simpler shape-only abstraction.

The **RandomELA** agent is an ablation where the RG speaker agent is not trained. We use it as a way to control for possible confounders. In both environment's barycentric plots, it shows an intrinsic bias towards the Shape-only language abstraction. This indicates that some aspects of the overall setting—potentially the environment, and/or the agent's architecture—inherently favor texture/shape over color in the MiniGrid environments. We assume that this favoring is mainly due to the environment. Indeed, in tasks like MultiRoom, door shapes must be identified regardless of color; and in KeyCorridor, the distinct shapes of (i) the key to find, (ii) followed by the uniquely-locked door to

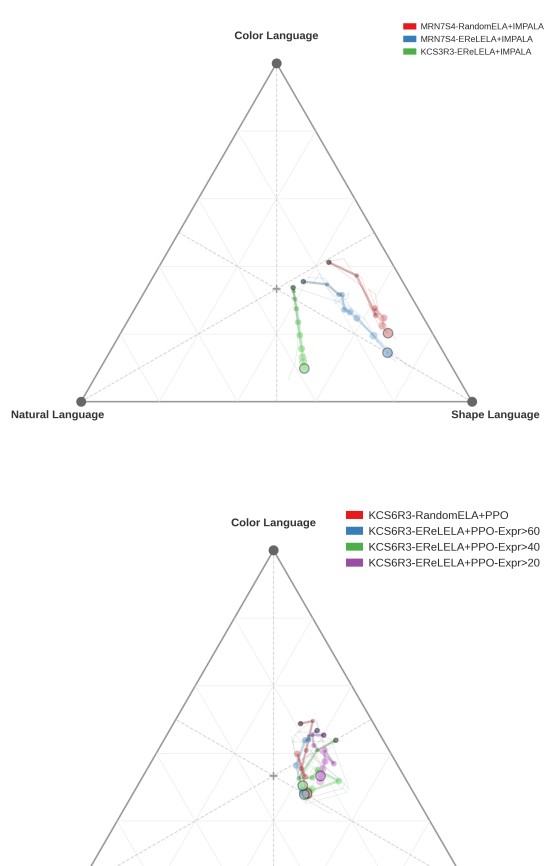

Figure 3: Compactness Ambiguity Metric (CAM) barycentric plots in *MultiRoom-N7-S4* (top) and *KeyCorridor-based* (bottom) environments from MiniGrid (Chevalier-Boisvert et al., 2023). The increase in the size of the barycenter dot represent the increase in RG training epoch.

unlock, and (iii) the ball to pickup are the primary features, once again being irrespective of their color.

When RG training is enabled, we observe a strong drift towards the Shape-only vertex as well, albeit different from one environment to the other, often converging in a region similar from the related **RandomELA** context. However, given that we can see clear RL performance differences between **RandomELA** and our method (see performance in KeyCorridor-S6R3 in Figure 2), this implies that subtle differences in our CAM barycentric plots have a strong impact onto the RL training. Thus, even small deviations from the **RandomELA**'s trajectories ought to be considered meaningful. Therefore, our result suggests that the resultant ELs do exploits the dominant, task-relevant features of the environment (shape), as opposed to anything else. However, the closeness to the **RandomELA** context may imply some amount of under-training with respect to exploring the full space of possible useful abstractions. It remains to be seen whether this is a limitation or a feature of the proposed method, and we will address this further in subsequent works.

**RQ3: How does the RG training impact the RL performance?** EReLELA is designed as an episodic, first-visit count-based method, akin to the strong baseline **FirstVisit-Count** (Henaff et al., 2023). However, **FirstVisit-Count** operates on the true symbolic observations, providing a near-oracle signal (with pseudo-relative expressivity of 100%), whereas EReLELA approximates state abstractions. As expected, EReLELA slightly underperforms the oracle (see KeyCorridor-S6R3 graph in Figure 2). We find that the **RandomELA** ablation, which employs discrete state abstractions but lacks the RG training, performs with a high standard error, and marginally worse than our approach. This validates the necessity of Referential Game (RG) training to structure the discrete abstractions. However, it performs better than our method when RG training is only performed until relative expressivity (RExpr) is higher than $20\%$. Moreover, we observe that performance systematically improves as the required RExpr threshold is increased from $20\%$ to $60\%$. This provides strong evidence that the quality of the ELs abstraction is directly constrained by the relative expressivity of the EL, and controlling this aspect is a critical element of the RG training, towards bringing EReLELA's performance closer to the upper-bound, oracle **FirstVisit-Count**'s performance.

## 5 CONCLUSION

In this work, we presented EReLELA, a novel exploration method that leverages emergent language abstractions to generate a stable, discrete, intra-life count-based novelty signal. Our expanded evaluation across four challenging environments, including the 3D **MiniWorld FullMazeS5**, showed that EReLELA is competitive with or exceeds recent state-of-the-art methods like ETD and RIDE, achieving SOTA performance in the FullMazeS5 task.

Our primary contribution lies in shifting the focus from simply using emergent communication to **controlling its abstractive quality**. We established that the **Relative Expressivity (RExpr)** of the emergent language is the critical variable, exhibiting a strong positive correlation with the agent's success rate ($R^2 \approx 0.52$), far superseding simpler training proxies like the number of distractors during RG training. Furthermore, our proposed **Compactness Ambiguity Metric (CAM)** revealed that emergent languages consistently learn a **minimal, task-relevant abstraction** (a Shape-only bias) rather than a compositional structure, efficiently latching onto the most sufficient features for RL success.

Moving forward, our results highlight the need to address the approximation gap between EReLELA and the symbolic oracle (**FirstVisit-Count**). Future research must focus on methods to deliberately control the abstraction bias—for instance, by enforcing compositional structures (driving the CAM barycenters towards the NL vertex) to enhance generalization. Testing this hypothesis in environments that emphasize **color-dominant features** will be crucial to verify the flexibility of the emergent language beyond the intrinsic Shape-only bias observed in MiniGrid.

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

## A    BROADER IMPACT

No technology is safe from being used for malicious purposes, which equally applies to our research. However, we view many of the ethical concerns surrounding research to be mitigated in the present case. These include data-related concerns such as fair use or issues surrounding use of human subjects, given that our data consists solely of simulations.

With regards to the ethical aspects related to its inclusion in the field of Artificial Intelligence, we argue that our work aims to have positive outcomes on the development of human-machine interfaces since we investigate, among other things, alignment of emergent languages with natural-like languages.

The current state of our work does not allow extrapolation towards negative outcomes. We believe that this work is of benefit to the research community of reinforcement learning, language emergence and grounding, in their current state.

# B    Further Experimental Details

Table 1: Summary of tested agent settings.

| Agent | RG Training | Observation Encoder Weights Sharing |
|---|---|---|
| *Synthetic Natural Language Abstraction* | N/A | N/A |
| *STGS-LazImpa-5-1 EReLELA (agnostic)* | LazImpa ($\beta_1 = 5, \beta_2 = 1$) | No |
| *STGS-LazImpa-10-1 EReLELA (shared)* | LazImpa ($\beta_1 = 10, \beta_2 = 1$) | Yes |
| *STGS-LazImpa-10-1 EReLELA (agnostic)* | LazImpa ($\beta_1 = 10, \beta_2 = 1$) | No |
| *Impatient-Only EReLELA (shared)* | Impatient-Only | Yes |
| *Impatient-Only EReLELA (agnostic)* | Impatient-Only | No |
| *RANDOM* | No | N/A |

**Synthetic Natural Language Oracles.** Like Tam et al. (2022), we employ language oracles that provides NL descriptions/captions of the state. Like them, we mean to use the adjective 'natural' to specify the quality and form of the caption rather than the process in which it is obtained (i.e. programmatically as opposed to having human beings producing them). Nevertheless, in order to make the distinction clear, we will refer to those oracles as Synthetic Natural Language (SNL) oracles.

We mean to emphasise that our considerations and results are agnostic to the process through which the NL captions are obtained, as we only indeed care about their quality and form, i.e. which vocabulary and grammar are being used, which here refers to that of the English natural language. We flag this as a limitation of our study because using NL captions produced from human beings would have yield a more varied and rich distribution, which would possibly impact the resulting RL agent's performance. We make the choice here to only use synthetically-generated NL captions because they can be generated "accurately and reliably, and at scale" (Tam et al., 2022).

Our implementation of SNL oracles are simply describing the visible objects in terms of their colour and shape attributes, from left to right on the agent's perspective, whilst also taking into account object occlusions. For instance, around the end of the trajectory presented in Figure 8, the green key would be occluded by the blue cube, therefore the SNL oracle would provide the description 'blue cube red cube' alone. We also implement colour-specific and shape-specific language oracles, which consists of filtering out from the SNL oracle's utterance the information that each of those language abstract away, e.g. removing any shape-related word in the colour-specific language.

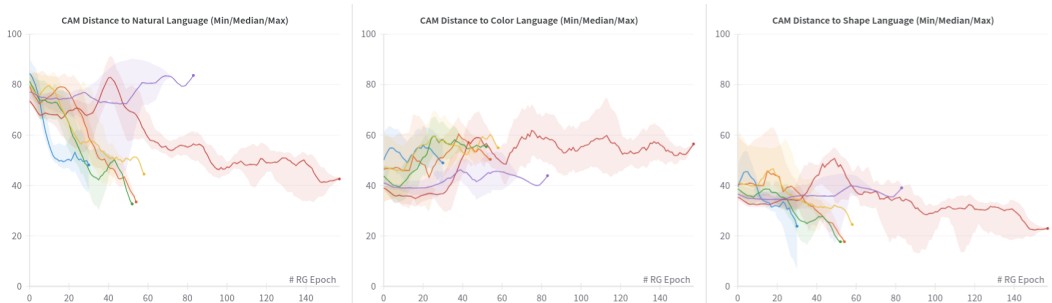

Figure 4: CAM distances to SNL (left), Color language (middle), and Shape language (right), for ELs brought about in *KeyCorridor-S3-R2* from MiniGrid (Chevalier-Boisvert et al., 2023), with different agents: (i) the *STGS-LazImpa-$\beta_1$-$\beta_2$ EReLELA* agents with $\beta_1 = 5$ (agnostic only) or $\beta_1 = 10$ (shared and agnostic), and $\beta_2 = 1$, (ii) the *Impatient-Only EReLELA* agents (shared and agnostic), and (iii) the *RANDOM* agent referring to an ablated version of EReLELA without RG training.

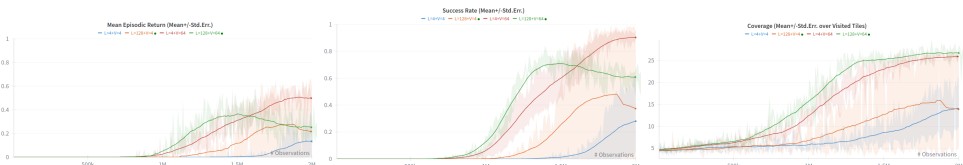

Figure 5: Mean episodic return (left), success rate learning curve (middle), and coverage learning curve (right) (32 running average steps), in *MultiRoom-N7S4* from MiniGrid (Chevalier-Boisvert et al., 2023), for different EReLELA agents trained with IMPALA in an *intra-life* context, with varying communication channel sizes.

## C   FURTHER BACKGROUND

### C.1   ON EXPLORATION TYPES IN RL

Stanton & Clune (2018) identifies two categories of exploration strategies, to wit *across-training*, where novelty of states, for instance, is evaluated in relation to all prior training RL episodes, and *intra-life*, where it is evaluated solely in relation to the current RL episode. Historically, we can identify two types of intrinsic motivation explorations depending on how the intrinsic reward is computed, either relying on count-based or prediction-based methods. Count-based methods estimate the frequency of an agent encountering a specific state (or similar states) by maintaining a "state pseudo-count," which quantifies novelty based on how rarely a state has been visited. Prediction-based methods, on the other hand, compute novelty by measuring an agent's uncertainty about predicting future observations. While prediction-based methods are typically used in across-training strategies (Pathak et al., 2017; Burda et al., 2018), count-based methods can be applied to both intra-life and across-training strategies, although they are more commonly extended to across-training approaches (Bellemare et al., 2016; Ostrovski et al., 2017) (cf. Appendix C.2 for more relevant details). Our proposed EReLELA architecture relies on an *intra-life* count-based method which can be extended as an across-training approach (cf. Section 3.1).

Finally, task-related nuance regarding the difficulty of the exploration task must be made; depending on whether the environment remains the same from one episode to the next (*singleton*) or changes from one episode to another, for instance by being procedurally generated. Exploration tasks involving procedurally-generated environments are referred to as hard-exploration tasks, and they are notoriously difficult for count-based exploration methods (Raileanu & Rocktäschel, 2019; Zha et al., 2021). Indeed, when states are procedurally-generated, almost all states will be showing 'novel' features, most times irrespectively of whether it is relevant to the task or not. It will follow that their state (pseudo-)count will always be low and therefore the RL agent will get feedback towards reaching all of them indefinitely, but if every state is 'novel' then there is nothing to guide the agent in any specific direction that would amount to good exploration.

### C.2   ON COUNT-BASED EXPLORATION METHODS IN RL

In the context of an intrinsic reward signal correlated with surprise, then it is necessary to quantify how much of surprise each observation/state provides. Intuitively, we can count how many times a given observation/state has been encountered and derive from that count our intrinsic reward. The reward would guide the RL agent to prefer rarely visited/observed states compared to common states. This is referred to as the count-based exploration method. Count-based exploration method were originally only applicable to tabular RL where the state space is discrete and it is easy to compare states together. When dealing with continuous or high-dimensional state spaces, such method is not practical. Thus, Bellemare et al. (2016) proposed (and extended in Ostrovski et al. (2017)) a pseudo-count approach which was derived from increasingly more efficient density models, and they showed success in applying it to image-based exploration environments from Atari 2600 benchmark, such as *Montezuma's Revenge*, *Private Eye*, and *Venture*.

Another approach to counting states from continuous and/or high-dimensional state spaces is by relying on hashing functions, so that states become tractable. Indeed, Tang et al. (2016) have shown that

a generalisation of classical counting techniques through hashing can provide an appropriate signal for exploration in continuous and/or high-dimensional environments where informed exploration is required. In effect, they proposed to discretise the state space $\mathcal{S}$ with a hash function $\phi : \mathcal{S} \to \mathbb{Z}^k$, with $k \in \mathbb{N} \setminus \{0\}$, to derive an exploration bonus of the form $r^+(s) = \frac{\beta}{\sqrt{n(\phi(s))}}$ where $\beta \in \mathbb{R}^+$ is a bonus coefficient and $n(.)$ is a count initialised at zero for the whole range of $\phi$ and updated at each step $t$ of the RL loop by increasing by 1 the count $n(\phi(s_t))$ related to the current observation/state $s_t$. Performance is dependent on the hash function $\phi$, and especially in terms of granularity of the discretisation it induces. Indeed, it would be desirable that the 'similar' states result in hashing collisions while the 'distant' states would not. To this end, they propose to use locality-sensitive hashing (LSH) such as SimHash (Charikar, 2002), resulting in the following:

$$\phi(s) = \text{sgn}(Ag(s)) \in \{-1, 1\}^k, \tag{5}$$

where sgn is the sign function, $A \in \mathbb{R}^{k \times D}$ is a matrix with each entry drawn i.i.d. from a standard Gaussian distribution, and $g : S \to \mathbb{R}^D$ is an optional preprocessing function. Note that increasing $k$ leads to higher granularity and therefore decreases the number of hashing collisions. Tang et al. (2016) reports great results on the Atari 2600 benchmarks, both with and without a learnable $g$ that is modelled as the encoder of an autoencoder (AE).

## C.3 ON EMERGENT COMMUNICATION

In RGs, sometime called Lewis' discrimination game (Rita et al., 2022), typically, the speaker is prompted with a target stimulus and the listener action is to discriminate it from some other distractor stimuli based solely on the message it observes from the speaker. Distractor stimuli are selected using a distractor sampling scheme, which can range from a simple uniform sampling from the set of all stimuli to more elaborated techniques taking into account the probabilities of observing target and distractor stimuli in real or imagine contexts. The distractor sampling scheme has been shown to impact the properties of the resulting EL (Lazaridou et al., 2016; 2018). Visual (discriminative) RGs have been shown to be well-suited for unsupervised representation learning, either by competing with state-of-the-art self-supervised learning approaches on downstream classification tasks (Dessì et al., 2021), or because they have been found to further some forms of disentanglement Higgins et al. (2018); Kim & Mnih (2018); Chen et al. (2018); Locatello et al. (2020) in learned representations (Xu et al., 2022; Denamganaï et al., 2023). Disentanglement can enable "better up-stream performance" (van Steenkiste et al., 2019), greater sample-efficiency, and some form of (systematic) generalization (Montero et al., 2021; Higgins et al.; Steenbrugge et al., 2018). Indeed, disentanglement is thought to reflect the compositional structure of the world, thus disentangled learned representations ought to enable an agent wielding them to generalize along those lines. The work of Chaabouni et al. (2020) showed that, in the context of generative, symbolic (i.e. disentangled stimuli) referential games, the degree of compositionality of the emerging languages and the agents ability to generalize to zero-shot stimuli are not correlated, but (i) "when a language is positionally disentangled (and, to a lesser extent, bag-of-symbols disentangled), it is very likely that the language will be able to generalize – a guarantee we do not have from less informative topographic similarity", and (ii) the data regime (e.g. low or high) is a better predictor for generalization (i.e. "generalization emerges 'naturally' if the input space if large").

# D  COMPUTATIONAL DETAILS OF THE COMPACTNESS AMBIGUITY METRIC

We recall from Section 3.2 that a CAM measure consists of a distribution, represented by an histogram of $N$ bins, where $N$ is one of the two hyperparameters of the metric. We refer to the counts in the bins as CAM scores. The CAM takes as inputs (i) a video-like input framed as a dataset of $N_{\mathcal{D}}$ RL trajectories of length $T$: $\mathcal{D} = \{s_t \in \mathcal{S} | t \in [1, T \cdot N_{\mathcal{D}}]\}$, and (ii) a speaker agent whose utterances are in the language $l$ that we want to evaluate with the metric. In order to formally define the speaker agent, we first define a language $l$ as a subset of $V^L$ where $V$ is a vocabulary with $|V|$ tokens and $L$ is the maximum length of each utterance/caption. Thus, for each language $l \subseteq V^L$, we define a speaker $\mathrm{Sp}_l : \mathcal{S} \mapsto V^L$, such that $\mathrm{Sp}_l(\mathcal{D}) = l$.

Next, we refer to the length of the time-interval that each utterance $u \in l$ occupies over dataset $\mathcal{D}$ (video input) as a compactness count of the said utterance. At each timestep $t$, if a caption $u_t = \mathrm{Sp}(s_t) \in l$ occurs and it differs from the one at $t - 1$, then a compactness count is associated to utterance $u_t$ (cf. lines 4-8 in Alg. 1).

---

**Algorithm 1:** Compactness Ambiguity Metric (CAM)

**Given**  :
- $\mathcal{D}$: Dataset of $N_{\mathcal{D}}$ RL trajectories of length $T$;
- $\mathrm{Sp}_l$: Speaker agent for language $l$ being evaluated;
- $N$: Number of histogram bins;
- $(\lambda_i)_{i \in \{0,1,\dots,N-1\}} \in [0,1]^N$: partition hyperparameters;

**Initialize** :
- $H \leftarrow \mathbf{0} \in \mathbb{R}^N$;
- $\mathcal{RA}_l(\mathcal{D}) \leftarrow \frac{|\mathcal{D}|}{\#\mathrm{Sp}_l(\mathcal{D})}$;
- $\forall i \in \{0, 1, \dots, N-1\}$ initialise $T_i$ with Eq. 7;

/* **Estimate compactness counts:**                                    */
1 $t_{\text{start}} \leftarrow 0$;
2 **foreach** $t, s_t \in enumerate(\mathcal{D})$ **do**
3      $u_t \leftarrow \mathrm{Sp}_l(s_t)$;
4      **if** $t > 0$ *and* $u_t \neq u_{t-1}$ **then**
5          $c \leftarrow t - t_{start}$;
6          $\delta_{\mathcal{D}}^l(u_{t-1}) \leftarrow \delta_{\mathcal{D}}^l(u_{t-1}) \cup \{c\}$;
7          $t_{\text{start}} \leftarrow t$;
8      **end**
9 **end**
/* Last state's regularisation:                                        */
10 $\delta_{\mathcal{D}}^l(u_{T \cdot N_{\mathcal{D}}-1}) \leftarrow \delta_{\mathcal{D}}^l(u_{T \cdot N_{\mathcal{D}}-1}) \cup \{T \cdot N\mathcal{D} - 1 - t_{\text{start}}\}$;
/* **Generate histogram:**                                             */
11 **foreach** $u \in Sp_l(\mathcal{D})$ **do**
12      **foreach** $c \in \delta_{\mathcal{D}}^l(u)$ **do**
13          Find bin index $i \in [0, N-1]$ s.t. $T_i \leq c < T_{i+1}$;
14          $H(i) \leftarrow H(i) + 1$;
15      **end**
16 **end**
**Output**  : $H$: Histogram of compactness counts;

---

This association is captured by a mapping from utterances $u \in l$ to sets of compactness counts. We denote it as the compactness count function defined as $\delta_{\mathcal{D}}^l : l \rightarrow 2^{\mathbb{N}}$ for language $l$ over dataset $\mathcal{D}$. In other words, for each $u \in l$ over $\mathcal{D}$, the set $\delta_{\mathcal{D}}^l(u)$ contains the numbers of consecutive timesteps for which $u$ was uttered by $\mathrm{Sp}_l$, without being uttered in the previous timestep. For instance, if we consider $u \in l$ such that the inverse function of the speaker $\mathrm{Sp}_l^{-1} : V^L \mapsto \mathcal{S}$ yields $\mathrm{Sp}_l^{-1}(u) = \{s_{t_1}, s_{t_1+1}, s_{t_1+2}, s_{t_2}\}$, with $(t_1, t_2) \in [0, T]^2$ such that $t_2 > t_1 + 3$, then $\delta_{\mathcal{D}}^l(u) = \{3, 1\} \in 2^{\mathbb{N}}$ because $u$ occurred 2 non-consecutive times over $\mathcal{D}$. Those non-consecutive

occurrences lasted for, respectively, 3 and 1 consecutive timesteps, which amounts to compactness counts of 3 and 1.

Next, we focus on the histogram that the metric returns. To sort compactness counts in this histogram, it is necessary to associate to each bin a partition of admissible compactness counts. Since compactness counts refer to time intervals, each bin of the histogram must refer to a range of time, between $0$ and the maximum length $T$ of an RL trajectory/episode in the given environment. We assume that the start of the range associated with a given bin is the end of the range associate with the previous bin. Therefore, we can naïvely associate to each bin $i \in \{0, 1, \ldots, N-1\}$ a time interval start $T_i$, defined relatively to the maximal length $T$. This framing is shown in Equation 6, with $\lceil \cdot \rceil$ being the ceiling operator. It is obtained by partitioning the whole range with the second and last hyperparameters $(\lambda_i)_{i \in \{0, 1, \ldots, N-1\}} \in [0, 1]^N$ such that $\forall (j, k),\ j < k \implies \lambda_j < \lambda_k$:

$$T_i = 1 + \lceil \lambda_i \cdot T \rceil \tag{6}$$

For regularisation purposes, we define $T_N = T$. Thus, by definition, bin $i \in 0, 1, \ldots, N-1$ will contain all the compactness counts $c$ belonging to the timespan $[T_i, T_{i+1}]$ (cf. lines 11-16 in Alg. 1).

In Appendix F.1, we show that this framing is sufficient to grant internal validity to our metric, meaning that this framing of the CAM (i) enables us to discriminate between different languages that are known to build different state-abstractions (e.g. synthetic languages that refers to all or only one specific attribute of objects, such as color or shape, used to caption a video stream that is egocentric viewpoint of an agent randomly walking in a 3D room with many randomly-placed objects), and (ii) maps languages without consistent state-abstractions (e.g. shuffled captions over a video stream) close to a null distribution histogram.

Despite this framing yielding internal validity, it is not optimal in our RL context. Indeed, we show in Appendix D.1 that this naïve framing is not only sensitive to abstractions performed by the language but also to redundancy in the dataset $\mathcal{D}$. Redundancy can occur in our RL-focused framing when $k \geq 2$ consecutive states are identical, for instance when the RL agent uses an action that does not affect its observations. These state-level redundancy situations artificially inflate compactness counts, which our metric captures as language abstractions whereas they are not. We show in Appendix D.1 that framing the bin's thresholds $T_i$ with respect to the relative ambiguity of the tested language, instead of the maximal length $T$ of an RL trajectory in the environment, yields greater sensitivity to abstractions and reduces the impact of redundancy onto the metric.

We define the relative ambiguity of a language $l$ as $\mathcal{RA}_l(\mathcal{D}) = \frac{|\mathcal{D}|}{\#\mathrm{Sp}_l(\mathcal{D})}$, where $|\cdot|$ being the size operator over collections (differing from sets in the sense that they allow duplicates, and the $|\cdot|$ operator accounting for them) and $\#$ the set cardinality operator. The framing based on relative-ambiguity is shown in Equation 7:

$$T_i = 1 + \lceil \lambda_i \cdot \mathcal{RA}_l(\mathcal{D}) \rceil \tag{7}$$

In the remainder of the paper, we report CAM measures using this framing.

### D.1 COMPARING FRAMEWORKS OF THE COMPACTNESS AMBIGUITY METRIC

We consider the ambiguity of a given language $l$, defined as $\mathcal{A}_l = \frac{\#\text{unique stimuli}}{\#\text{unique utterances}}$ with $\#$ the set cardinality operator. Dealing with stimuli being states of a (randomly-walking) RL agent, gathered into a dataset $\mathcal{D}$, the number of unique states or stimuli cannot be estimated reliably when dealing with complex, continuous stimuli. Thus, the best we can rely on is a measure of relative ambiguity over a dataset, that we define as $\mathcal{RA}_l(\mathcal{D}) = \frac{\#\text{stimuli}}{\#\text{unique utterances}} = \frac{|\mathcal{D}|}{\#\mathrm{Sp}_l(\mathcal{D})}$, with $|\cdot|$ being the size operator over collections (differing from sets in the sense that they allow duplicates). In those terms, the relative ambiguity is minimized if and only if (i) $\#\mathcal{D} = |\mathcal{D}|$, and (ii) $\mathrm{Sp}_l$ is injective. On the other hand, considering that a language $l$ performs an abstraction over $\mathcal{D}$ is tantamount to some stimuli $(s, s') \in \mathcal{D}^2$ sharing the same utterance $u = \mathrm{Sp}_l(s) = \mathrm{Sp}_l(s')$, i.e. consisting of a hash collision, meaning that the mapping $\mathrm{Sp}_l$ from $\mathcal{D}$ to $l$ would not be injective and therefore $\mathrm{Sp}_l$ would not be bijective.

Incidentally, the relative ambiguity $\mathcal{RA}_l(\mathcal{D})$ cannot be minimized, leading to the language $l$ being ambiguous over $\mathcal{D}$. In this consideration, we can see that the ambiguity of a language (over a given dataset) can be impacted by either the extent to which an abstraction is performed (meaning that

most colliding states occur on consecutive timesteps) or the extent to which the dataset is redundant, with many duplicate states which may or may not be consecutive (meaning $\#\mathcal{D} << |\mathcal{D}|$). This allows us to identify two possibly sources of ambiguity. Therefore, in order to build a metric that measures abstractions' qualities, it is important to focus on sources of ambiguities that are the result of consecutive-timesteps states colliding, more than sources of ambiguities that are the result of redundancy in the given dataset.

Thus, we propose to build the CAM in a way that minimises its sensibility to redundancy-induced ambiguity. This is achieved at the level of the timespan-focused buckets. Indeed, for a given language $l$ and dataset $\mathcal{D}$, we define the buckets' related timespans in relation to the relative ambiguity $\mathcal{RA}_l(\mathcal{D}) = \frac{1}{\mathcal{RE}_l(\mathcal{D})} = \frac{|\mathcal{D}|}{\#\mathrm{Sp}_l(\mathcal{D})}$, as shown in Equation 8 with $\lambda_i \in [0,1]$ $s.t.$ $\forall(j,k),\ j < k \implies \lambda_j < \lambda_k$, and $\lceil \cdot \rceil$ being the ceiling operator. This is in lieu of naïve definition in relation to the maximal length $T$ of an episode in the environment, as shown in Equation 9.

$$\forall i \in [0, N-1],\ T_i = 1 + \lceil \lambda_i \cdot \mathcal{RA}_l(\mathcal{D}) \rceil \tag{8}$$

$$\forall i \in [0, N-1],\ T_i' = 1 + \lceil \lambda_i \cdot T \rceil \tag{9}$$

$$\forall i \in [0, N-1],\ CA(l, \mathcal{D})_{T_i} = \sum_{u \in l} \frac{\#\mathbb{I}(\delta_{\mathcal{D}}^l(u) \geq T_i)}{\#\delta_{\mathcal{D}}^l(u)} \tag{10}$$

More formally, let us first acknowledge decomposition of relative ambiguity over two independent quantities, one for each of its sources being either abstraction or redundancy, such that $\mathcal{RA}_l = \mathcal{RA}_l^{\text{redundancy}} + \mathcal{RA}_l^{\text{abstract}}$. Then note that the relative ambiguity is equal to the mean number of consecutive timesteps, or compactness count, for which a given utterance would be used when the unique utterances are uniformly distributed over the dataset $\mathcal{D}$. Thus, in the metric, we propose to absorb variations of relative ambiguity due to redundancy by changing the metric's bucket setup, from Equation 9 to Equation 8. Doing so, it is true that the metric's bucket setup will also vary when the abstraction-induced relative ambiguity varies, we remark that the metric would not build invariant to this source of relative ambiguity since it is taken into accounts when sorting out the different unique utterances into their relevant bucket, based on the maximal number of consecutive timesteps in which they occur. This mechanism is shown in equation 10 where $\delta_{\mathcal{D}}^l : l \to 2^{\mathbb{N}}$ is the compactness count function that associates each utterances $u \in l$ to its related set of compactness counts over dataset $\mathcal{D}$, i.e. the set that contains numbers of consecutive timesteps for which $u \in l$ was uttered by $\mathrm{Sp}_l$, each time it was uttered without being uttered in the previous timestep. For instance, recall that if we consider $u \in l$ such that $\mathrm{Sp}_l^{-1}(u) = \{s_{t_1}, s_{t_1+1}, s_{t_1+2}, s_{t_2}\}$, with $(t_1, t_2) \in [0, T]^2$ such that $t_2 > t_1 + 3$, then $\delta_{\mathrm{D}}(u) = \{3, 1\}$ because $u$ occurred 2 non-consecutive times over $\mathcal{D}$ and those occurrences lasted for, respectively, 3 and 1 consecutive timesteps, i.e. for compactness counts of 3 and 1. The indicator function $\mathbb{I}(\cdot)$ along with $\geq T_i$ in $\mathbb{I}(\delta_{\mathcal{D}}^l(u) \geq T_i)$ implies filtering of the output set based on compactness counts being greater or equal to $T_i$. We provide in appendix D.2 an analysis of the sensitivity of our proposed metric, and in appendix F.1 experimental results that ascertain the internal validity of our proposed metric, we consider a 3D room environment of MiniWorld (Chevalier-Boisvert et al., 2023), filled with 5 different, randomly-placed objects (cf. Figure 8).

## D.2 SENSITIVITY ANALISYS OF THE COMPACTNESS AMBIGUITY METRIC

Based on derivative-based local sensitivity analysis, we propose an intuitive proof of our claim that defining timespans in relation to the relative ambiguity reduces the sensibility to variations induced by redundancy-based ambiguity in the resulting metric, compared to defining timespans in relation to the the maximal length $T$ of an agent's trajectory in the environment. To do so, we assume:

(i) that there exists two differentiable function $f_i.f_i'$ such that for all $i \in [1, N]$, we have $CA(\mathcal{D})_{T_i} = f_i(\mathcal{D}, \mathcal{RA}_l^{\text{redundancy}}, \mathcal{RA}_l^{\text{abstract}})$ when $T_i$ is defined according to Equation 7, and respectively with $f_i'$ when using $T_i'$ from Equation 6, and

(ii) that their partial derivatives with respect to $T_i$ or $T_i'$ are negative. Indeed, $T_i$ and $T_i'$ are involved into filtering operations reducing the value of the numerator in Equation **??**, therefore any increase of their values would result in decreasing the overall metric output, which implies that their partial derivatives with $f_i$ and $f_i'$ must be negative.

With those assumptions, we show that $f_i$'s sensitivity to redundancy-induced ambiguity $\mathcal{RA}_l^{\text{redundancy}}$ is less than that of $f_i'$:

*Proof.*

$$\frac{\partial f_i}{\partial \mathcal{RA}_l^{\text{redundancy}}} = \frac{\partial f_i}{\partial CC_\mathcal{D}} \cdot \frac{\partial CC_\mathcal{D}}{\partial \mathcal{RA}_l^{\text{redundancy}}} + \frac{\partial f_i}{\partial T_i} \cdot \frac{\partial T_i}{\partial \mathcal{RA}_l^{\text{redundancy}}}$$

(from Assump. (i) about $f_i$)

$$\iff \frac{\partial f_i}{\partial \mathcal{RA}_l^{\text{redundancy}}} = \frac{\partial f_i'}{\partial \mathcal{RA}_l^{\text{redundancy}}} + \frac{\partial f_i}{\partial T_i} \cdot \frac{\partial T_i}{\partial \mathcal{RA}_l^{\text{redundancy}}}$$

(from Assump. (i) about $f_i'$)

$$\iff \frac{\partial f_i}{\partial \mathcal{RA}_l^{\text{redundancy}}} = \frac{\partial f_i'}{\partial \mathcal{RA}_l^{\text{redundancy}}} + \frac{\partial f_i}{\partial T_i} \cdot \lambda_i$$

$$\implies |\frac{\partial f_i}{\partial \mathcal{RA}_l^{\text{redundancy}}}| \leq |\frac{\partial f_i'}{\partial \mathcal{RA}_l^{\text{redundancy}}}|$$

(since $\frac{\partial f_i}{\partial T_i} \cdot \lambda_i \leq 0$ from Assump. (ii))

$\square$

# E PRELIMINARY EXPERIMENTS

## E.1 IMPACT OF REFERENTIAL GAME ACCURACY

In this experiments, we investigate whether the RG accuracy impacts the RL agent training, in the context of the *MultiRoom-N7-S4* environment from *MiniGrid* (Chevalier-Boisvert et al., 2023), with an RL sampling budget of $1M$ pixel-based observations.

**Hypothesis.** We seek to validate the following hypotheses, **(PH1)** : the sample-efficiency of the RL agent is dependant on the quality of the RG players, as parameterised by the $acc_{RG-thresh}$ hyperparameter.

**Evaluation.** We report both the success rate and the coverage count in the hard-exploration task of *MultiRoom-N7-S4*. To compute the coverage count, we overlay a grid of tiles over the environment's possible locations/cells of the agents and we count the number of different tiles visited by the RL agent over the course of each episode. We use 3 random seeds for each agent. In order to evaluate the impact of the RG accuracy strictly in terms of the kind of abstractions that are being performed by the resulting EL, we use the *Impatient-Only* loss function (removing the impact of the hyperparameter of the scheduling function $\alpha(\cdot)$ from the *Lazy* term of the *STGS-LazImpa* loss function), and we employ an **agnostic** version of our proposed EReLELA agent, i.e. **without sharing the observation encoder between the RG players and the RL agent**. We present results for two different RG accuracy threshold $acc_{RG-thresh} = 60\%$ (green) or $acc_{RG-thresh} = 80\%$ (red), and compare against, as an upper bound the *Natural Language Abstraction* agent (blue), which refers to using the NL oracle to compute intrinsic reward, and, as a lower bound an ablated version of EReLELA without RG training (orange).

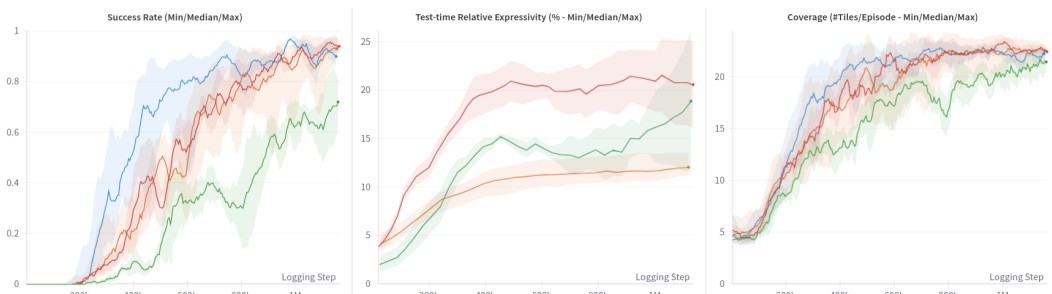

Figure 6: Success rate (left), test-time relative expressivity (middle), and per-episode coverage count (right) in *MultiRoom-N7-S4* from MiniGrid (Chevalier-Boisvert et al., 2023), computed as running averages over 256 episodes each time (i.e. 32 in parallel, as there are 32 actors, over 8 running average steps), for different agents: (i) the *Natural Language Abstraction* agent (blue) refers to using the NL oracle to compute intrinsic reward, the *Agnostic Impatient-Only EReLELA* agent refers to our proposed architecture **without sharing the observation encoder between the RG players and the RL agent**, using the Impatient-Only loss function to optimize the RG players, with an RG accuracy threshold $acc_{RG-thresh} = 60\%$ (ii - green) or $acc_{RG-thresh} = 80\%$ (iii - red), and (iv) an ablated version without RG training (orange).

**Results.** We present results in Figure 6. We observe statistically significant differences between the performances (in terms of success rate, cf. Figure 6(left)) of the two EReLELA agents with $acc_{RG-thresh} = 60\%$ or $acc_{RG-thresh} = 80\%$, thus validating hypothesis (PH1). We observe that higher RG accuracy threshold lead to higher sample-efficiency.

As a sanity check, we plot the results of the ablated EReLELA agent without RG training, and we were expecting it to perform poorer than any other agent since the quality of its RG players is the lowest, at chance level. Yet, we observe that it performs on par with the best $acc_{RG-thresh} = 80\%$-EReLELA agent. While puzzling, we propose a possible explanation in the observation that the test-time relative expressivity of the ablated agent is higher than that of the least-performing, $acc_{RG-thresh} = 60\%$-EReLELA agent, and on par with that of the best-performing, $acc_{RG-thresh} = 80\%$-EReLELA agent, at the beginning of the RL agent training process. Thus, we interpret this as follows: the randomly-initialised ablated agent's EL is possibly performing an abstraction over the observation

space that is good-enough for the RL agent to start learning exploration skills, the same way the random network in the context of the RND agent from Burda et al. (2018) probably does, and increasing the quality of the RG players may only be a sufficient condition to increasing the sample-efficiency of the EL-guided RL agent.

### E.2 IMPACT OF REFERENTIAL GAME DISTRACTORS

In this experiments, we investigate whether the RG's number of distractors $K$ and distractor sampling scheme impacts the RL agent training, in the context of the *KeyCorridor-S3-R2* environment from *MiniGrid* (Chevalier-Boisvert et al., 2023), with an RL sampling budget of $1M$ observations.

**Hypothesis.** We seek to validate the following hypotheses, **(PH2)** : the sample-efficiency of the RL agent is dependant on the number of distractors $K$ and the distractor sampling scheme.

**Evaluation.** We report the success rate in the hard-exploration task of *KeyCorridor-S3-R2*. We use 3 random seeds for each agent. Like previously, we use the *Impatient-Only* loss function (to remove the impact of the hyperparameter of the scheduling function $\alpha(\cdot)$ from the *Lazy* term of the *STGS-LazImpa* loss function), and we employ an **agnostic** version of our proposed EReLELA agent, i.e. **without sharing the observation encoder between the RG players and the RL agent**. We present results for three different number of distractors $K \in [15, 128, 256]$ and two different sampling scheme between *UnifDSS* corresponding to uniformly sampling distractors over the whole training dataset, or *Sim50DSS* corresponding to sampling distractors $50\%$ of the time from the same RL episode than the current target stimulus is from and, the rest of the time following *UnifDSS*. Following results in Appendix E.1, we set the RG accuracy threshold $acc_{RG-thresh} \in [80\%, 90\%]$.

**Results.** We present results in Figure 7. We observe statistically significant differences between the performances of the different EReLELA agents, thus validating hypothesis (PH2). Our results show that (i) the number of distractors $K$ is the most impactful parameter and it correlates positively with the resulting performance, irrespective of the distractor sampling scheme used, and, indeed, (ii) while the *Sim50DSS* seems to provide better performance than *UnifDSS* for low numbers of distractors $K = 15$, although not statistically-significantly, the table is turned when considering high number of distractors $K = 256$ where the *UnifDSS* yields statistically significantly better performance than the *Sim50DSS*.

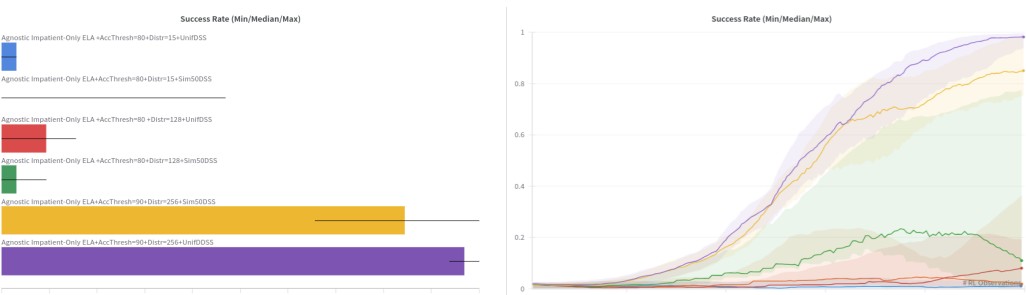

Figure 7: Final success rate barplot (left) and success rate throughout learning (right) in *KeyCorridor-S3-R2* from MiniGrid (Chevalier-Boisvert et al., 2023), computed as running averages over 1024 episodes each time (i.e. 32 in parallel, as there are 32 actors, over 32 running average steps), for the *Agnostic Impatient-Only EReLELA* agent, which refers to our proposed architecture **without sharing the observation encoder between the RG players and the RL agent**, using the Impatient-Only loss function to optimize the RG players, with different number of distractors $K$ and distractors sampling schemes: with RG accuracy threshold $acc_{RG-thresh} = 80\%$, (i) $K = 15$ and *UnifDSS* or Sim50DSS, (ii) $K = 1128$ and *UnifDSS* or Sim50DSS, or with RG accuracy threshold $acc_{RG-thresh} = 90\%$, (iii) $K = 256$ and *UnifDSS* or Sim50DSS.

# F FURTHER EXPERIMENTS

## F.1 EXPERIMENT #1: INTERNAL VALIDITY OF THE COMPACTNESS AMBIGUITY METRIC

**Environment.** We consider a 3D room environment of MiniWorld (Chevalier-Boisvert et al., 2023), where the agent's observation is egocentric, as a first-person viewpoint. The room is filled with 5 different, randomly-placed objects, with different shapes (among ball, box or key) and colours (among). The dimensions simulate a 12 by 5 meters room, like shown in a top-view perspective in Figure 8.

**Hypothesis.** In this experiments, we seek to validate two hypotheses, **(H1.1)** : the Compactness Ambiguity Metric captures something that is related to the kind of abstraction a language performs, and **(H1.2)** : the Compactness Ambiguity Metric allows a graduated comparison of different kind of abstractions being performed, meaning that it allows discrimination between different kind of abstractions.

**Evaluation.** In order to compute the metric, we use 5 seeds to gather random walk trajectories in our environment, for each language. In order to evaluate (H1.1), we propose to measure a language that is built to present no meaningful abstractions and we expect the measure to be close to null. We build a language that performs no meaningful abstraction from the natural language oracles by shuffling its utterances over the set of agent trajectories that are used to compute the metric, meaning that the mapping between temporally-sensitive stimuli and linguistic utterances is rendered completely random.

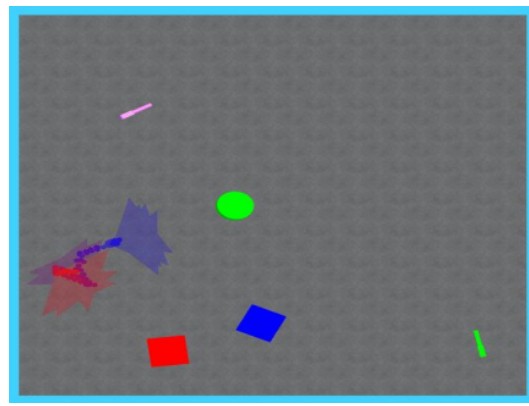

Figure 8: Top-view visualization of a wall-free 3D environment with different objects (e.g. red and blue cubes, purple and green keys, and green ball) showing the trajectory (from blue to red dots) of a randomly-walking embodied agent, with first-person perspectives highlighted at relevant timesteps using colored cones - showing the agent's viewpoint direction when a new utterance is used to describe the first-person perspective using an oracle speaking in NL.

Then, in order to evaluate (H1.2), we show experimental evidences that the metric allows qualitative discrimination between the different languages built above from the natural language oracles, which are build to perform different kind of abstractions.

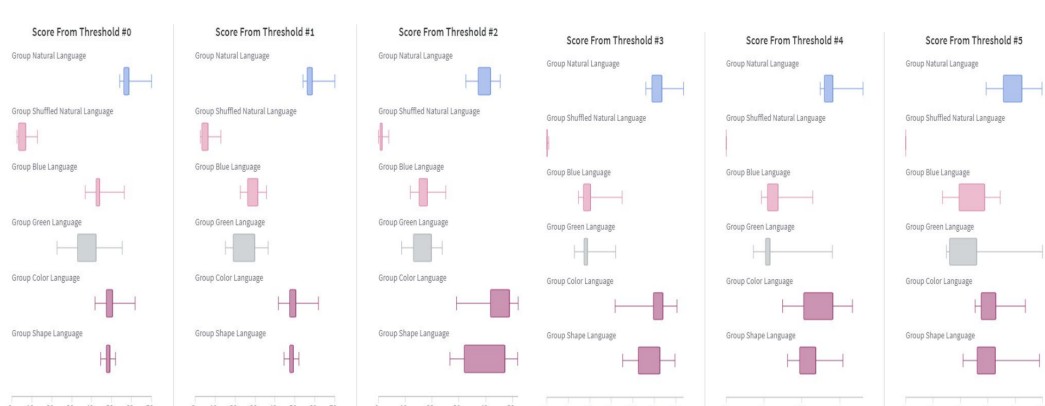

Figure 9: Interval validity measures of Compactness Ambiguity Metric for $N = 6$ timespans/thresholds, with $\lambda_0 = 0.0306125$, $\lambda_1 = 0.06125$, $\lambda_2 = 0.125$, $\lambda_3 = 0.25$, $\lambda_4 = 0.5$ and $\lambda_5 = 0.75$, for different languages built to perform different kind of abstraction. We can qualitatively discriminate between each languages, and validate that the shuffled (natural) language's meaningless abstraction scores almost null.

**Results.** We present results of the metric with $N = 6$ timespans in Figure 9, for $\lambda_0 = 0.0306125$, $\lambda_1 = 0.06125$, $\lambda_2 = 0.125$, $\lambda_3 = 0.25$, $\lambda_4 = 0.5$ and $\lambda_5 = 0.75$. As the shuffled (natural) language measure is almost null on all timespans/thresholds, we validate hypothesis (H1.1).

We observe that we can qualitatively discriminate between each evaluated language's measures since the histograms are statistically different. Moreover, language abstractions scores are inversely correlated with the amount of information being abstracted away, i.e. attribute-value-specific languages' abstraction score lower than colour/shape-specific languages abstraction, which score lower than natural language abstractions. Thus, we can see that the metric is graduated and that the graduation follows the amount of abstraction being performed by each language. This allows us to validate hypothesis (H1.2).

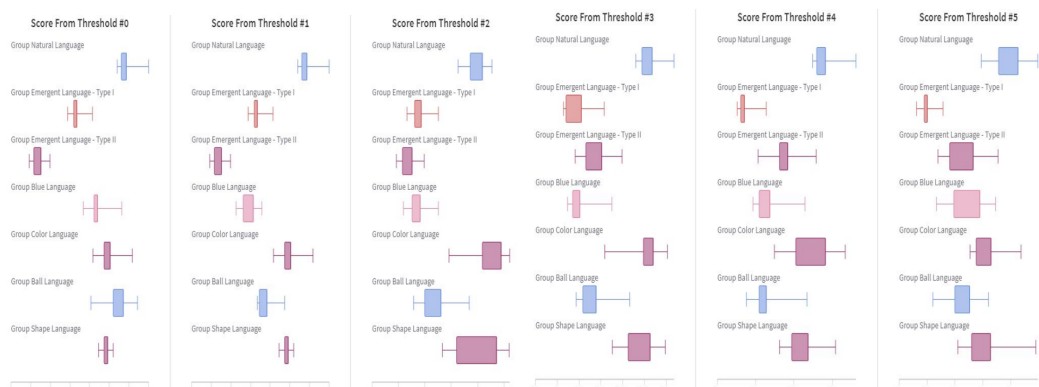

Figure 10: Measures of Compactness Ambiguity Metric for $N = 6$ timespans/thresholds, with $\lambda_0 = 0.0306125$, $\lambda_1 = 0.06125$, $\lambda_2 = 0.125$, $\lambda_3 = 0.25$, $\lambda_4 = 0.5$ and $\lambda_5 = 0.75$, comparing ELs (Type I and II) with different oracles' languages built to perform different kind of abstraction.

### F.2 EXPERIMENT #2: QUALITIES OF EMERGENT LANGUAGES ABSTRACTIONS IN 3D ENVIRONMENT

In this experiment, we investigate what kind of abstractions do ELs perform over a 3D environment, in comparison to some natural languages abstractions, as detailed at the beginning of Section 4. For further precision, we also implement attribute-value-specific language oracles with the same filtering approach. For instance, for the green value on the colour attribute, we would obtain a green-only language oracle whose utterances could be 'EoS' if no visible object is green, or 'green green' if there are two green objects visible in the agent's observation. We consider the same 3D room environment of MiniWorld (Chevalier-Boisvert et al., 2023) as in Section F.1, i.e. the agent's observation is egocentric, as a first-person viewpoint and the room is filled with 5 different, randomly-placed objects, with different shapes (among ball, box or key) and colours (among). The dimensions simulate a 12 by 5 meters room, like shown in a top-view perspective in Figure 8.

**Hypothesis.** We seek to validate the following hypotheses, **(H2.1)** : ELs build meaningful abstractions, and **(H2.2)** : ELs brought about using the STGS-LazImpa loss function (type II) perform more meaningful abstractions than Impatient-Only baseline (type I).

**Evaluation.** In order to make the CAM measures, we use 5 seeds to gather random walk trajectories in our environment, for each language. In order to evaluate both (H2.1) and (H2.2), we use the CAM to measure the kind of abstractions performed by ELs brought about in the two different EReLELA settings, with Impatient-Only or STGS-LazImpa losses, and compare those measures with those of the oracles' languages that we previously studied.

**Results.** We present results of the metric with $N = 6$ timespans in Figure 10. We observe statistically significant differences between ELs of type I and II, with type I's abstraction being similar to a Blue-specific language's abstraction (timespans $0 - 4$) or a Ball-specific language's abstraction (timespans $1 - 3$), and type II's abstraction not really resembling any of the oracle languages' abstractions, but still being meaningful with scores increasing along with the length of the considered timespans. Thus,

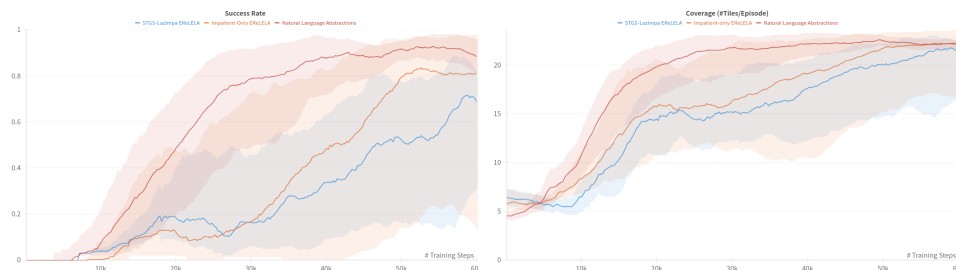

Figure 11: Success rate (left) and per-episode coverage count (right) in *MultiRoom-N7-S4* from MiniGrid (Chevalier-Boisvert et al., 2023), computed as running averages over 1024 episodes each time (i.e. 32 in parallel, as there are 32 actors, over 32 running average steps), for different agents: (i) the *Natural Language Abstraction* agent (NLA) refers to using the NL oracle to compute intrinsic reward, (ii) the *STGS-LazImpa EReLELA* agent refers to our proposed architecture, EReLELA, using the STGS-LazImpa loss function to optimize the RG players, and (iii) the *Impatient-Only EReLELA* agent refers to the same architecture without the lazy-speaker loss to optimize the RG players.

we validate hypothesis (H2.1), but cannot conclude on hypothesis (H2.2), unless we consider that CAM scores related to longer timespans are more meaningful, for instance.

### F.3 EXPERIMENT #3: LEARNING PURELY-NAVIGATIONAL SYSTEMATIC EXPLORATION SKILLS FROM SCRATCH

In the following, we present an experiment in the *MultiRoom-N7-S4* environment from *Mini-Grid* (Chevalier-Boisvert et al., 2023), which is possibly less challenging than *KeyCorridor-S3-R2*, presented in the Section 4, for it does not involve as many complex object manipulation (e.g. only open/close doors, no unlocking of doors – which requires the corresponding key to be firstly picked up – nor pickup/drop keys or other objects as distractors), but still poses a **purely-navigational** hard-exploration challenge. We report results on the **agnostic** version of our proposed EReLELA architecture, that is to say **without sharing the observation encoder between both RG players and the RL agent**, in order to guard ourselves against the impact of possible confounders found in multi-task optimization, such as possible interference between the RL-objective-induced gradients and the RG-training-induced gradients. We use an RG accuracy threshold $acc_{RG-thresh} = 65\%$ and a number of training distractors $K = 3$ (like at testing/validation time).

**Hypotheses.** We consider whether NL abstractions can help for a purely-navigational hard-exploration task in RL with a count-based approach **(H3.0)**, and refer to the relevant agent using NL abstractions to compute intrinsic rewards as NLA. Then, we make the hypothesis that ELs can be used similarly **(H3.1)**, and we investigate to what extent do ELs compare to NLs in terms of abstraction performed, in this purely-navigational task. In the case of (H3.1) being verified, we would expect ELs to perform similar abstractions as NLs **(H3.2)**.

**Evaluation.** We evaluate (H3.0) and (H3.1) using both the success rate and the coverage count. To compute the coverage count, we overlay a grid of tiles over the environment's possible locations/cells of the agents and we count the number of different tiles visited by the RL agent over the course of each episode. To evaluate (H3.2), we compute the CAM scores of both the ELs and the oracles' natural, color-specific, and shape-specific languages. As we remarked that an agent's skillfullness at the task would induce very different trajectories (e.g. in *MultiRoom-N7-S4*, staying in the first room and only ever seeing the first door, for an unskillfull agent, as opposed to visiting multiple rooms and observing multiple colored-doors, for a skillfull agent), we compute the oracle languages CAM scores on the exact same trajectories than used to compute each EL's CAM scores.

**Results.** We present in Figure 11(left) the success rate of the different agents, and the per-episode coverage count in Figure 11(right). From the fact that both the NLA and EReLELA agent performance converges higher or close to 80% of success rate, we validate hypotheses (H0) and (H3.1), in the context of the *MultiRoom-N7-S4* environment. We remark that the sample-efficiency is slightly better for NLA than it is for EL-based agents, possibly because of the fact that ELs are learned online in parallel of the RL training, as opposed to the case of NLA which makes use of a ready-to-use

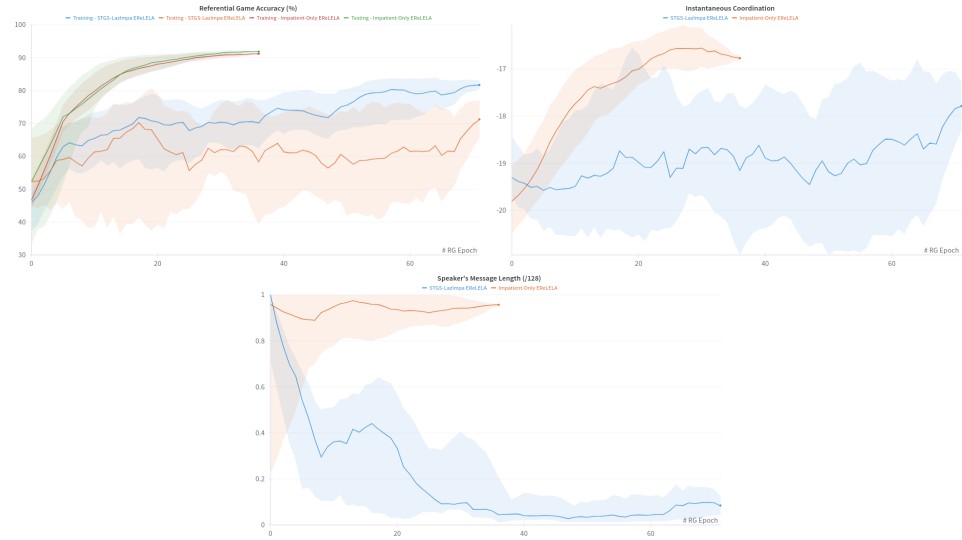

Figure 12: Performance and qualities of the ELs brought about in the context of both (i) the *STGS-LazImpa EReLELA* agent, and (ii) the *Impatient-Only EReLELA* agent, with respect to both the training- and validation/testing-time RG accuracy (left), the validation/test-time Instantaneous Coordination (Jaques et al., 2018; Lowe et al., 2019; Eccles et al., 2019)(middle), and the validation/testing-time length of the speaker's messages (as a ratio over the max sentence length $L = 128$ - right).

oracle. Among the two EReLELA agents, the learning curves are not statistically-significantly distinguishable, meaning that learning systematic exploration skills with EReLELA can be done with some robustness to the anecdotical differences in qualities of the different ELs due to using different optimization losses. Indeed, we also report in Figure 12 both the training- and validation/testing-time RG accuracies (on the left), the validation/testing-time Instantaneous Coordination (in the middle – Jaques et al. (2018); Lowe et al. (2019); Eccles et al. (2019)), and the validation/testing-time length of the RG speaker's messages (on the right), showing that the ELs brought about in the two different contexts perform differently in terms of their RG objective and have different qualities, but these discrepancies do not seem to impact the RL agents learning equally well from the different abstractions they perform (as evidenced in the next paragraph).

Next, with regards to hypothesis (H3.2), we investigate whether the two contexts bring about ELs that perform different abstractions, and how do these relate to the abstractions performed by natural, colour-specific, and shape-specific languages, by showing in Figure 13 their CAM scores. We observe that both contexts result in ELs performing abstractions similar or better than colour-specific languages, which is to be expected as (door) colours are the most salient features of the environment. Indeed, the only two shapes or objects visible are 'wall' and 'door', whereas there are more than 7 different colours of interest. In the context of the Impatient-Only EReLELA agent, the EL's abstractions are scoring very similarly to NL abstractions, as we consider longer timespans (from timespans #2 to #5).

We could hypothesise that without the lazy-ness constraint the speaker agent may be given enough capacity to compress/express information pertaining to the location of visible objects, as this information is the only one that is captured by the NL oracle but not captured by the shape- and colour-specific languages.

### F.4   EXPERIMENT #4: TOWARDS QUANTIFYING RL AGENTS' LEARNING PROGRESS

In the context of RGs, the speed at which a language emerges (in terms of sampled observations, or number of games played) may possibly remain constant, when the data and the player architectures are fixed. Thus, when the data changes, the rate of language emergence may change too. Incidentally, we are entitled to ponder whether some properties of the data, which here are RL trajectories, would influence the rate of language emergence and how?

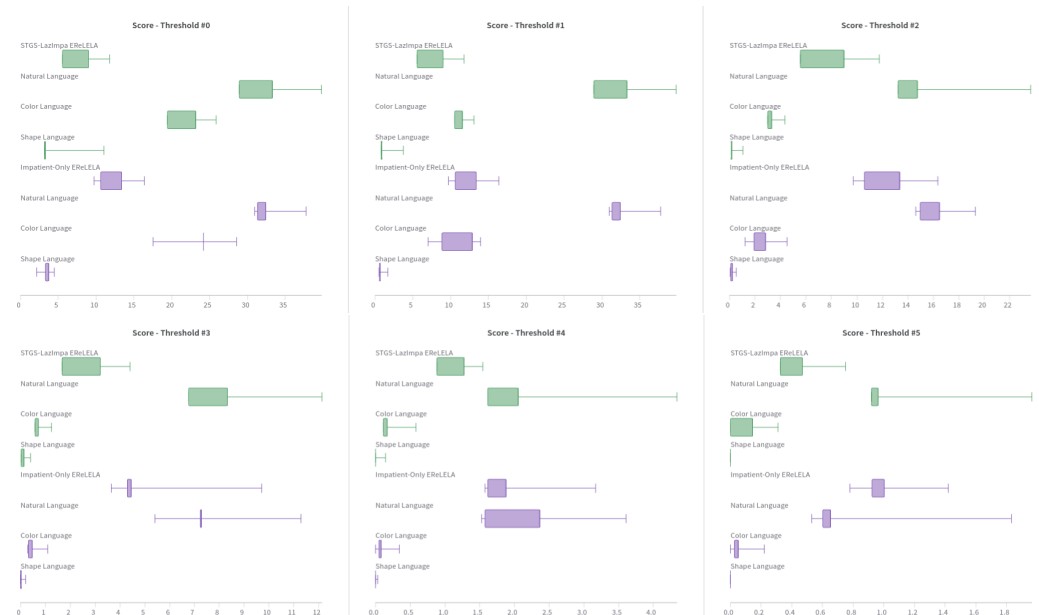

Figure 13: Comparison of Compactness Ambiguity Metric scores for $N = 6$ timespans/thresholds, with $\lambda_0 = 0.0306125$, $\lambda_1 = 0.06125$, $\lambda_2 = 0.125$, $\lambda_3 = 0.25$, $\lambda_4 = 0.5$ and $\lambda_5 = 0.75$, between the abstractions performed by ELs brought about in the context of both (i) the *STGS-LazImpa EReLELA* agent (in green, first rows) and (ii) the *Impatient-Only EReLELA* agent (in purple, bottom rows), and the abstractions performed by the natural, colour-specific, and shape-specific languages, computed on the very same agent trajectories.

**Hypothesis.** We hypothesise that as the RL agent gets more skillful, the expressivity of the emergent language increases **(H4.1)**. Indeed, at each RG training epoch, the size of the dataset is fixed, and as the stimuli gets more diverse when the RL agent gets more skillful at exploring, the RG training will prompt the EL to increase its expressivity.

**Evaluation.** To verify our hypothesis, we propose to measure the skillfullness of the RL agent in terms of exploration using the per-episode coverage count metric, and we measure the expressivity of the EL via the test-time (Relative) Expressivity after each RG training epoch.

**Results.** We present results in Figure 14, that show the (relative) expressivity of the ELs does exhibit variations throughout the learning process of the RL agent. And, if we perform a regression analysis with each runs in terms of the per-episode coverage count of the RL agent on the x-axis and the expressivity of the ELs on the y-axis, we obtain a high coefficient of determination between the two metrics, $R^2 = 0.4642$. Thus, we conclude that the (relative) expressivity of the ELs in EReLELA can provide a way to quantify the progress of the RL agent, at least when it comes to exploration skills.

**Limitations.** Exploration skills translates directly into diversity of the stimuli being observed, and therefore it prompts any RG players to increase the expressivity of their communication protocol, but it is remains to be seen whether this effect is valid in any environment. For instance, it is unclear whether a skillfull player in any other video game would induce the same effect on the diversity of the stimuli encountered. Thus, it is worth investigating whether this correlation holds for other genre of environments and skills, which we leave to future works.

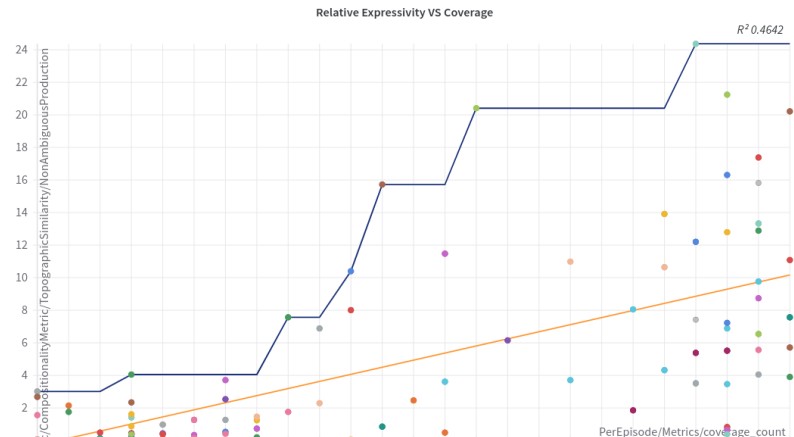

Figure 14: Relative expressivity of the EL as a function of the per-episode coverage of the RL agent, at the end of training, over multiple runs with different hyperparameters during a W&B Sweep (Biewald, 2020).

### F.5 ABLATION STUDIES

**Agents.** In the following, our agents are optimized using the R2D2 algorithm (Kapturowski et al., 2018) with the Adam optimizer Kingma & Ba (2014). We use $\lambda_{int} = 0.1$ and $\lambda_{ext} = 10.0$ in order to make sure that the agent pursues the external goal once the exploration of the environment has highlighted it. Further details about the RL agent can be found in Appendix G. For our RG agents, we consider optimization using either the Impatient-Only or the STGS-LazImpa loss function from Rita et al. (2020), but the latter is adapted to the context of a Straight-Through Gumbel-Softmax (STGS) communication channel (Havrylov & Titov, 2017; Denamganaï & Walker, 2020c). We refer to it as STGS-LazImpa. The details of the loss including the two hyperparameters $\beta_1, \beta_2$ can be found in Appendix H.1. Regarding the LazImpa (Rita et al., 2020) loss function, it has been shown to induce Zipf's Law of Abbreviation (ZLA) in the ELs. Thus, we can investigate in the following experiments how does **structural** similarity between NLs and ELs affect the kind of abstractions they perform, as well as the resulting RL agent. Further details about the RG in EReLELA can be found in Appendix H. A summary of tested agent settings is presented in Table 1 of Appendix B.

**Environments & Comparison Agents.** After having considered in our preliminary experiments with R2D2 (cf. Appendix F.4) the 2D environment *MultiRoom-N7-S4*, we propose below experiments in the more challenging *KeyCorridor-S3-R2* environment from MiniGrid (Chevalier-Boisvert et al., 2023). Indeed, it involves complex object manipulations, such as (distractors) object pickup/drop and door unlocking, which requires first picking up the relevantly-colored key object.

Like Tam et al. (2022), we employ language oracles that provides NL descriptions/captions of the state, that we use as a strong upperbound (under the assumption that the NL-performed abstractions are the gold-standard we could have). We discuss it further in Appendix B.

**Hypotheses.** We seek to validate the following hypotheses. Firstly, we consider whether a simple count-based approach over (synthetic) NL abstractions is sufficient to solve hard-exploration RL tasks **(H1)**. We refer to the corresponding agent using (synthetic) NL abstractions to compute intrinsic rewards as SNLA. We carry on with the hypothesis that a simple count-based approach over EL abstractions is similarly sufficient **(H2)**. In doing so, we will also investigate to what extent do ELs compare to SNLs in terms of abstractions, using our proposed CAM. Using our proposed CAM, we consider two state abstractions to be aligned when their CAM distance is low. As the *MultiRoom-N7-S4* environment only shows differently-coloured doors in a partial observation context, the most important type of state abstraction is related to the colour of visible objects. On the other hand, since the *KeyCorridor-S3-R2* environment requires picking up an object behind a (unique) locked door, after having unlocked said door with a key, the most important type of state abstraction is related to the shape of visible objects. We consider a state abstraction to be meaningful in a given environment if it is aligned with the language oracle's abstraction that is the most important in said environment. Thus, we expect ELs to perform meaningful abstractions **(H3)**, i.e. being aligned with

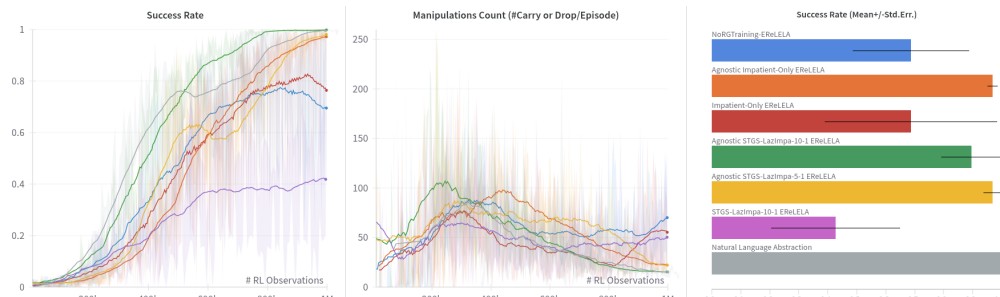

Figure 15: Success rate learning curve (left), computed as running averages over 1024 episodes each time (i.e. 32 in parallel, as there are 32 actors, over 32 running average steps), and barplot (right), along with per-episode manipulation count (middle) in *KeyCorridor-S3-R2* from MiniGrid (Chevalier-Boisvert et al., 2023), for different agents: (i) the *Natural Language Abstraction* agent (SNLA) refers to using the SNL oracle to compute intrinsic reward, (ii) the *STGS-LazImpa-$\beta_1$-$\beta_2$ EReLELA* agents with $\beta_1 = 5$ (agnostic only) or $\beta_1 = 10$ (shared and agnostic), and $\beta_2 = 1$, (iii) the *Impatient-Only EReLELA* agents (shared and agnostic), and (iv) the *RANDOM* agent referring to an ablated version of EReLELA without RG training.

the colour-specific language's abstractions in the *MultiRoom-N7-S4* environment, and being aligned with the shape-specific language's abstractions in the *KeyCorridor-S3-R2* environment.

**Evaluation.** We employ 3 random seeds for each agent (due to time complexity and 'GPU-poor'-ness). We evaluate (H1) and (H2) using both the success rate and the manipulation count, in the hard-exploration task of *KeyCorridor-S3-R2*. The manipulation count is a per-episode counter incremented each time an object is successfully picked up or dropped by the RL agent over the course of each episode. In order to evaluate (H3), we use the CAM to measure the kind of abstractions performed by ELs, and compare those measures with those of the oracles' languages that we previously detailed.

**RG Communication Channel Ablation.** We consider the parameters of the RG communication channel, here parameterised with the STGS. We denote $L$ and $V$ to be, respectively, the maximum sentence length and the vocabulary size of the channel. Results are shown in Figure 5 of Appendix B. The main trend is that overcomplete-ness of the communication channel is helpful, and more so when it is thanks to a high vocabulary size rather than a high maximum sentence length.

**EReLELA learns Systematic Navigational & Manipulative Exploration Skills from Scratch.**

We present in Figure 15 both the success rate of the different agents (as line plot through learning -left-, or barplot at the end of learning -right-), and the per-episode manipulation count (middle). We observe that both the SNLA and EReLELA agent performances converge higher or close to 80% of success rate (except the STGS-LazImpa-10-1). These results mean that it is possible to learn systematic exploration skills from either of SNL or EL abstractions with a simple count-based exploration method, in 2D environments (cf. further evidence in Appendix E.1 with the *MultiRoom-S7-R4* environment). We therefore validate hypotheses (H1) and (H2), and remark that they put into perspective the directions of previous literature designing complex exploration algorithms (Burda et al., 2018; Badia et al., 2019), compared to that of designing more efficient state abstraction methods.

The sample-efficiency is better for SNLA than it is for most EL-based agents, except the Agnostic STGS-LazImpa-10-1 agent, possibly because of the fact that ELs are learned online in parallel of the RL training, as opposed to the case of SNLA which makes use of a ready-to-use oracle. Concerning the most-sample-efficient Agnostic STGS-LazImpa-10-1 agent, we interpret its success to be the result of benefiting from both a language structure ascribing to the ZLA and a performed abstraction that is more optimal than SNL oracle's ones, because it is learned from the stimuli themselves. Among the different Agnostic EReLELA agents, the final performance are not statistically-significantly distinguishable, meaning that learning systematic exploration skills with EReLELA can be done with some robustness to the anecdotical differences in qualities of the different ELs. On the other hand, the shared/non-agnostic EReLELA agents's performance are statistically-significantly distinguishable from each other and from their agnostic versions, achieving lower performance. We suspect possible interferences between the RG training and the RL training, preventing any valuable representations from being learned in the shared observation encoder (cf. Figure 1). We will investigate in future works whether a synergy can be achieved. Acknowledging the RANDOM agent, which is the

ablated version of EReLELA without RG training, enabling still a median performance around 70% of success rate, we recall the RND approach (Burda et al., 2018), for they both share a randomly initialised networked from which feedback is harvested to guide an RL agent. This ablated version is not a lower-bound but rather an interesting ablation that enables us to show the impact of the RG training, increasing the sample-efficiency and final performance of the RL agent.

**EReLELA learns Meaningful Abstractions.** Regarding hypothesis (H3), we show in Figure 4 of Appendix B the CAM distances between the different agent's ELs and the natural, colour-specific, and shape-specific languages. We recall that in the *KeyCorridor-S3-R2* environment, the most important feature is object shape as the agent must pickup a key from all other distractor objects and then use it to unlock the locked door. Thus, as we observe that most ELs' abstractions are closer to the shape-specific language than the others, we conclude that EReLELA learns meaningful abstractions, thus validating hypothesis (H3) (cf. Appendix F.5 for further evidence in the context of *MultiRoom-N7-S4*). Further, we remark that the failing STGS-LazImpa-10-1 EReLELA agent is indeed failing because its EL's abstractions are not highlighting shape features. When considering the shared/non-agnostic agents only, we can see that they require many more RG training epochs, meaning that they reach the accuracy threshold less often than their agnostic counterparts. We take this as further evidence for our interpretation that there might be interference between the RL objective and the RG objective. We note that abstractions from ELs brought about in the contexts of the *Agnostic STGS-LazImpa* agents and the *Agnostic Impatient-Only* agents are the closest to that of the shape-specific language ones, and their evolution throughout learning are similar. Yet, the *Agnostic STGS-LazImpa* agents achieves statistically-significantly better sample-efficiency (cf. Figure 7). We interpret this as being caused by the ZLA structure of the ELs in the context of the *Agnostic STGS-LazImpa* agents, thus showing that NL-like structure is impacting the performed abstractions in ways that are yet to be unveiled.

# G AGENT ARCHITECTURE

The EReLELA architecture is made up of three differentiable agents, the language-conditioned RL agent and the two RG agents (speaker and listener). Each agent contains at least a visual/observation encoder module that can be shared between agents.Both RG agents contain a language module that is not shared. The *listener* agent additionally incorporates a third decision module that combines the outputs of the other two modules. The RL agent similarly incorporates a third decision module with the addition that this third module contains a recurrent network, acting as core memory module for the agent. Using the Straight-Through Gumbel-Softmax (STGS) approach in the communication channel of the RG, the *speaker* agent is prompted to produce the output string of symbols with a *Start-of-Sentence* symbol and the visual module's output as an initial hidden state while the *listener* agent consumes the string of symbols with the null vector as the initial hidden state. In the following subsections, we detail each module architecture in depth.

**Visual Module.** The visual module $f(\cdot)$ consists of the *Shared Observation Encoder*, which can be shared between all the different agents.The former consists of three blocks of convolutional layers of sizes $8, 4, 3$ with strides $4, 3, 1$, each followed by a 2D batch normalization layer and a ReLU non-linear activation function. The two first convolutional layers have 32 filters, whilst the last one has $64$. The bias parameters of the convolutional layers are not used, as it is common when using batch normalisation layers. Inputs are stimuli consisting of RGB frames of the environment resized to $64 \times 64$.

**Language Module.** The language module $g(\cdot)$ consists of some learned Embedding followed by either a one-layer GRU network (Cho et al., 2014) in the case of the RL agent, or a one-layer LSTM network (Hochreiter & Schmidhuber, 1997) in the case of the RG agents. In the context of the *listener* agent, the input message $m = (m_i)_{i \in [1,L]}$ (produced by the *speaker* agent) is represented as a string of one-hot encoded vectors of dimension $|V|$ and embedded in an embedding space of dimension 64 via a learned Embedding. The output of the *listener* agent's language module, $g^l(\cdot)$, is the last hidden state of the RNN layer, $h_L^l = g^L(m_L, h_{L-1}^l)$. In the context of the *speaker* agent's language module $g^S(\cdot)$, the output is the message $m = (m_i)_{i \in [1,L]}$ consisting of one-hot encoded vectors of dimension $|V|$, which are sampled using the STGS approach from a categorical distribution $Cat(p_i)$ where $p_i = Softmax(\nu(h_i^s))$, provided $\nu$ is an affine transformation and $h_i^s = g^s(m_{i-1}, h_{i-1}^s)$. $h_0^s = f(s_t)$ is the output of the visual module, given the target stimulus $s_t$.

**Decision Module.** From the RL agent to the RG's listener agent, the decision module are very different since their outputs are either, respectively, in the action space $\mathcal{A}$ or the space of distributions over $K + 1$ stimuli (i.e. discriminating between distractors and target stimuli). For the RL agent, the decision module takes as input a concatenated vector comprising the output of visual module, after it has been procesed by a 3-layer fully-connected network with 256, 128 and 64 hidden units with ReLU non-linear activation functions, and some other information relevant to the RL context (e.g. previous reward and previous action selected, following the recipe in Kapturowski et al. (2018)). The resulting concatenated vector is then fed to the core memory module, a one-layer LSTM network (Hochreiter & Schmidhuber, 1997) with $1024$ hidden units, which feeds into the advantage and value heads of a 1-layer dueling network (Wang et al., 2016).

Regarding optimization of the RL agent, Table 2 highlights the hyperparameters used for the off-policy RL algorithm, R2D2(Kapturowski et al., 2018). More details can be found, for reproducibility purposes, in our open-source implementation at HIDDEN-FOR-REVIEW-PURPOSES.

Each run can be done on less than 2Gb of VRAM, and the amount of training time for a run, with e.g. one NVIDIA GTX1080 Ti, is between 24 and 48 hours depending on the architecture (e.g. shared or agnostic).

Table 2: Hyper-parameter values relevant to R2D2 in the EReLELA architecture presented. All missing parameters follow the ones in Ape-X (Horgan et al., 2018).

| R2D2 | |
| --- | --- |
| Number of actors | 32 |
| Actor update interval | 1 env. step |
| Sequence unroll length | 20 |
| Sequence length overlap | 10 |
| Sequence burn-in length | 10 |
| N-steps return | 3 |
| Replay buffer size | $1 \times 10^4$ obs. |
| Priority exponent | 0.9 |
| Importance sampling exponent | 0.6 |
| Discount $\gamma$ | 0.98 |
| Minibatch size | 64 |
| Optimizer | Adam (Kingma & Ba, 2014) |
| Learning rate | $6.25 \times 10^{-5}$ |
| Adam $\epsilon$ | $10^{-12}$ |
| Target network update interval | 2500 updates |
| Value function rescaling | None |

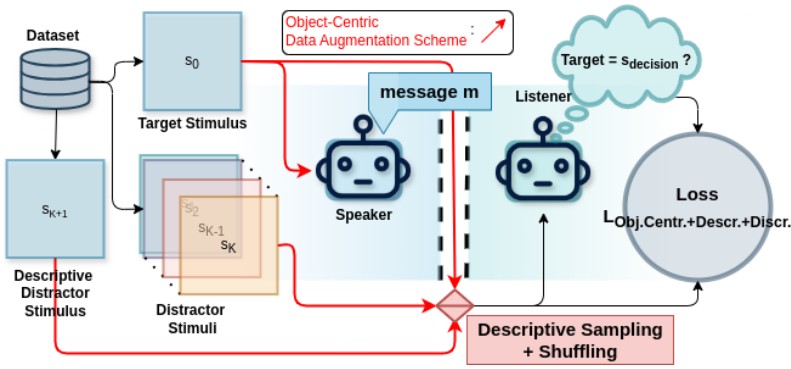

Figure 16: Illustration of a *descriptive object-centric (partially-observable) 2-players/L = 10-signal/N = 0-round/K-distractor* Referential Game variant, following the nomenclature from Denamganaï & Walker (2020b). Object-centrism is achieved via data augmentation schemes that are applied on to each stimulus before being fed to the different agents. As a $N = 0$-round variant, the Speaker agent only sends one message to the listener who cannot communicate back to, for instance ask questions. Based on this single message, the listener must be able to identify the target stimulus from the set of shuffled stimuli it receives, if it is present, or else specify that it is not present. Indeed, as a descriptive variant, the descriptive sampling can substitute the target stimulus for a descriptive distractor stimulus at a given frequency, in order to apply an extra pressure onto the listener agent.

## H ON THE REFERENTIAL GAME IN ERELELA

As detailed in Section 3.1, we focus on a *descriptive object-centric (partially-observable) 2-players/L = 10-signal/N = 0-round/K-distractor* RG variant (Denamganaï & Walker, 2020b), as illustrated in Figure 16.

We follow baseline implementation of the RG's listener from Havrylov & Titov (2017), i.e. the decision module builds a probability distribution over a set of $K + 1$ stimuli/images $(s_0, ..., s_K)$, consisting of $K$ distractor stimuli and the target stimulus, provided in a random order, given a message

$m$ using the scalar product:

$$p((d_i)_{i\in[0,K]}|(s_i)_{i\in[0,K]}; m) = Softmax\Big((h_L^l \cdot f(s_i)^T)_{i\in[0,K]}\Big). \tag{11}$$

However, our setting consist of a descriptive variant, on top of being discriminative. The descriptiveness implies that the target stimulus may not be passed to the listener agent, but instead replaced with a descriptive distractor. In effect, the listener agent's decision module therefore outputs a $K+2$-logit distribution where the $K+2$-th logit represents the meaning/prediction that a descriptive distractor has been introduced and none of the $K+1$ stimuli is the target stimulus that the speaker agent was 'talking' about. The addition is made following Denamganaï et al. (2023) as a learnable logit value, $logit_{no-target}$, it is an extra parameter of the model. Thus, in our case, the decision module output is no longer as specified in Equation 11, but rather as follows:

$$p((d_i)_{i\in[0,K+1]}|(s_i)_{i\in[0,K]}; m) = Softmax\Big((h_L^l \cdot f(s_i)^T)_{i\in[0,K]} \cup \{logit_{no-target}\}\Big). \tag{12}$$

The object-centrism is achieved via application of data augmentation schemes before feeding stimuli to any RG agent, following Dessì et al. (2021) but using Gaussian Blur transformation alone, as it was found sufficient in practice. We optimize the RG agents with either the Impatient-Only STGS loss and the STGS-LazImpa loss.

In the remainder of this section, we detail the STGS-LazImpa loss that we employed to optimize the referential game agents.

### H.1 STGS-LazImpa Loss

Emergent languages rarely bears the core properties of natural languages (Kottur et al., 2017; Bouchacourt & Baroni, 2018; Lazaridou et al., 2018; Chaabouni et al., 2020), such as Zipf's law of Abbreviation (ZLA). In the context of natural languages, this is an empirical law which states that the more frequent a word is, the shorter it tends to be (Zipf, 2016; Strauss et al., 2007). Rita et al. (2020) proposed LazImpa in order to make emergent languages follow ZLA.

To do so, Lazimpa adds to the speaker and listener agents some constraints to make the speaker lazy and the listener impatient. Thus, denoting those constraints as $\mathcal{L}_{STGS-lazy}$ and $\mathcal{L}_{impatient}$, we obtain the STGS-LazImpa loss as follows:

$$\mathcal{L}_{STGS-LazImpa}(m, (s_i)_{i\in[0,K]}) = \mathcal{L}_{STGS-lazy}(m) + \mathcal{L}_{impatient}(m, (s_i)_{i\in[0,K]}). \tag{13}$$

In the following, we detail those two constraints.

**Lazy Speaker.** The Lazy Speaker agent has the same architecture as common speakers. The 'Laziness' is originally implemented as a cost on the length of the message $m$ directly applied to the loss, of the following form:

$$\mathcal{L}_{lazy}(m) = \alpha(acc) \cdot |m| \tag{14}$$

where $acc$ represents the current accuracy estimates of the referential games being played, and $\alpha$ is a scheduling function as follows: $\alpha : accuracy \in [0,1] \mapsto \frac{accuracy^{\beta_1}}{\beta_2}$, with $(\beta_1, \beta_2) = (45, 10)$. It is aimed to adaptively penalize depending on the message length. Since the lazyness loss is not differentiable, they ought to employ a REINFORCE-based algorithm for the purpose of credit assignement of the speaker agent.

In this work, we use the STGS communication channel, which has been shown to be more sample-efficient than REINFORCE-based algorithms (Havrylov & Titov, 2017), but it requires the loss functions to be differentiable. Therefore, we modify the lazyness loss by taking inspiration from the variational autoencoders (VAE) literature (Kingma & Welling, 2013).

The length of the speaker's message is controlled by the appearance of the EoS token, wherever it appears during the message generation process that is where the message is complete and its length is fixed. Symbols of the message at each position are sampled from a distribution over all the tokens in the vocabulary that the listener agent outputs. Let $(W_l)$ be this distribution over all

tokens $w \in V$ at position $l \in [1, L]$, such that $\forall l \in [1, L]$, $m_l \sim (W_l)$. We devise the lazyness loss as a Kullbach-Leibler divergence $D_{KL}(\cdot|\cdot)$ between these distribution and the distribution $(W_{EoS})$ which attributes all its weight on the EoS token. Thus, we dissuade the listener agent from outputting distributions over tokens that deviate too much from the EoS-focused distribution $(W_{EoS})$, at each position $l$ with varying coefficients $\beta(l)$. The coefficient function $\beta : [1, L] \to \mathbb{R}$ must be monotically increasing. We obtain our STGS-lazyness loss as follows:

$$\mathcal{L}_{STGS-lazy}(m) = \alpha(acc) \cdot \sum_{l \in [1,L]} \beta(l) D_{KL}\Big((W_{EoS})|(W_l)\Big) \tag{15}$$

**Impatient Listener.** Our implementation of the Impatient Listener agent follows the original work of Rita et al. (2020): it is designed to guess the target stimulus as soon as possible, rather than solely upon reading the EoS token at the end of the speaker's message $m$. Thus, following Equation 11, the Impatient Listener agent outputs a probability distribution over a set of $K + 1$ stimuli $(s_0, ..., s_K)$ for all sub-parts/prefixes of the message $m = (m_1, ..., m_l)_{l \in [1,L]} = (m_{\leq l})_{l \in [1,L]}$ :

$$\forall l \in [1, L], \quad p((\mathbf{d_i^{\leq l}})_{\mathbf{i} \in [\mathbf{0,K}]}|(s_i)_{i \in [0,K]}; \mathbf{m}^{\leq \mathbf{l}}) = Softmax\Big((\mathbf{h}_{\leq \mathbf{l}} \cdot f(s_i)^T)_{i \in [0,K]}\Big), \tag{16}$$

where $\mathbf{h}_{\leq \mathbf{l}}$ is the hidden state/output of the recurrent network in the language module after consuming tokens of the message from position 1 to position $l$ included.

Thus, we obtain a sequence of $L$ probability distributions, which can each be contrasted, using the loss of the user's choice, against the target distribution $(D_{target})$ attributing all its weights on the decision $d_{target}$ where the target stimulus was presented to the listener agent. Here, we employ Havrylov & Titov (2017)'s Hinge loss. Denoting it as $\mathbb{L}(\cdot)$, we obtain the impatient loss as follows:

$$\mathcal{L}_{impatient/\mathbb{L}}(m, (s_i)_{i \in [0,K]}) = \frac{1}{L} \sum_{l \in [1,L]} \mathbb{L}((d_{i \in [0,K]}^{\leq l}, (D_{target})). \tag{17}$$