# OpenReview forum: "EReLELA: Exploration in Reinforcement Learning via Emergent Language Abstractions"
_ICLR.cc/2026/Conference — Submitted to ICLR 2026_

### Official Review · Reviewer_5gbL · 2025-10-26

**Soundness:** 2
**Presentation:** 1
**Contribution:** 2
**Rating:** 2
**Confidence:** 3

**Summary:**

The paper proposes an exploration algorithm based on language abstractions. One of the limitations of language abstraction–based exploration is the assumption of access to a state captioning system that takes environment observations (images, in this case) and produces captions, which are expensive to collect in practice. These captions serve as the language abstractions of the images. The paper proposes using emergent language systems trained in an unsupervised fashion to learn environment-specific language abstractions. The emergent language abstractions are then utilized for constructing intrinsic rewards and learning.
The paper also suggests using count-based algorithms instead of methods such as RND and NGU, arguing that count-based methods are simpler.
Finally, the paper proposes a metric for comparing different language abstractions and uses this metric to compare emergent language abstractions with natural language. It shows that the proposed algorithm learns emergent language abstractions that focus on the salient features with respect to the RL agent’s task, and it is similar to natural language-based abstractions.

**Strengths:**

- The idea of using EC for learning language tailored to the agent’s task is really appealing and interesting, and it can lift the burden of finding or designing an image captioning system.

- There is a good amount of analysis of the algorithm’s components, which is very helpful for understanding the importance of certain design choices

**Weaknesses:**

- Insufficient Evaluation: The algorithm has been evaluated on only two environments from MiniGrid. In one of these environments, it was outperformed by one of the baselines (although it performed better in the early stages of training). This undermines the effectiveness of the proposed method and does not provide sufficient evidence of its overall performance. I suggest running the method on more environments from MiniGrid with varying levels of difficulty, and then on more complex environments such as Craftax-Classic or Craftax.
- The paper is hard to follow and the presentation is relatively poor.

**Questions:**

- How many random seeds did the author use for their experiments in section 4 (before the ablation study, for figure 2)?
- In general, the experimental section is hard to follow, especially the ablation sections. (1) The plots are difficult to read without legends or labels on the learning curves, and the font size is quite small. (2) The ablation section still feels unstructured and hard to follow — it would benefit from some reorganization.(3)The paper overall might also benefit from restructuring to make it easier to read and follow.
- Some runs in the plots in Figure 4 are incomplete. I wonder whether the same trend continues for those incomplete runs.

I am willing to raise my score if the authors run their method on more environments and improve the clarity and structure of the paper. I really like the idea, but the paper currently feels unfinished and not ready for publication.

---

### Official Review · Reviewer_wjMv · 2025-10-28

**Soundness:** 2
**Presentation:** 1
**Contribution:** 3
**Rating:** 4
**Confidence:** 3

**Summary:**

This paper proposes EReLELA, a method that uses Emergent Language (EL) for exploration in Reinforcement Learning (RL). Through a referential game, it learns in an unsupervised way a state representation that is then used with a simple count-based exploration. The authors show that this approach improves exploration in sparse-reward MiniGrid environments compared to non-linguistic exploration-based baselines such as RND and RIDE. In addition, they propose further analyses showing that count-based methods are effective in general when combined with linguistic state representations, and that the learned ELs are aligned with ground-truth linguistic representations.

**Strengths:**

- Demonstrates that emergent languages (ELs) are a promising approach to enhancing exploration in reinforcement learning (RL), especially when combined with simple count-based methods.

- Achieves competitive results on standard benchmarks (MiniGrid) using a relatively simple method.

**Weaknesses:**

- Only three environments are tested, all from the same benchmark (MiniGrid). It would be important to include additional environments from other benchmarks to assess the generality of the approach.
- The paper is not self-contained. Too many key details are deferred to the appendix or to other papers, which makes the main text difficult to follow.
    - For instance, the use of CAM for H3 is shown only in a figure in the appendix.
- Unclear motivation for changing the RL algorithm. The paper switches between IMPALA and R2D2 without justification. Since EReLELA is supposed to be a general wrapper, it would be better to keep one base algorithm or explain the switch. As a result, it is unclear whether differences come from EReLELA or from the change in RL algorithm, making the ablations harder to interpret.
- The experimental section (particularly Section 4.1) is hard to follow. It mixes hypotheses, experiment descriptions, and results in a disorganized way. A cleaner structure would help. More specifically:
    - Section 4.1 mixes hypotheses, experimental descriptions, and results in an unorganized manner. The authors start by describing three hypotheses and experiments (Hypotheses), then interrupt with another experiment (channel ablation) that already includes results and analysis, and then return to the initial hypotheses with more results. It would be clearer to first present all hypotheses and experimental setups, and only afterward report and discuss the corresponding results.
    - Contradictory passages: for instance, the paragraph introducing the hypotheses states that H2 will use CAM, but in the evaluation paragraph it is H3 that uses CAM.
- The hypotheses themselves are somewhat confusing:
    - H1: Its relevance is unclear, as the experiment just above already shows that a simple count-based approach works. What is new here? It seems that the authors aim to show that count-based exploration also work with synthetic natural language abstractions (SNLA), but this is not clearly stated.
- The link between hypotheses and experiments is weak:
    - H2 is supposed to concern the relationship between SNL and EL abstractions, but the paper merely states “we therefore validate hypotheses (H1) and (H2)” without explaining how the results actually support H2.

Minor:
- L391 — **SNLA acronym not defined** when first introduced. It should be expanded to “Synthetic Natural Language Abstraction” upon first use.
- L448–L451 — **Unclear sentence:**
  “Acknowledging the RANDOM agent, which is the ablated version of EReLELA without RG training, enabling still a median performance around 70% of success rate, we recall the RND approach (Burda et al., 2018), for they both share a randomly initialised networked from which feedback is harvested to guide an RL agent.”
  → Sentence is overly long and unclear; especially the clause “enabling still…” and “networked” (should be “network”).
- L451 — **Typo:** “networked” → “network”.
- L309 — **Ambiguous reference:** “with symbolic observations (as opposed to pixel-based observations in the rest of the paper)” — specify where pixel-based results are reported.
- L254 — **Typo:** “maintian” → “maintain”.
- **Inconsistent spelling:** “normalisation” vs “normalization” — unify to one variant.
ve.

**Questions:**

1. Why didn’t you use the EL representation as part of the RL agent’s observation? Do you think incorporating it could improve performance?
2. NL-based state representations are useful not only for exploration but also for generalization. Do you think your method could help with generalization to new environments? Have you tested this?

---

### Official Review · Reviewer_Jzmy · 2025-10-30

**Soundness:** 2
**Presentation:** 1
**Contribution:** 2
**Rating:** 4
**Confidence:** 3

**Summary:**

This paper introduces EReLELA, an AI agent that leverages emergent language (so called EL, which is learned through unsupervised referential games) as an alternative to natural language (NL) for reinforcement learning (RL) in instruction-following tasks. Unlike NL, which is costly and limited, EL provides cheap and readily-available state abstractions. The study shows that EL-based agents can achieve comparable performance to NL-based in hard-exploration environments, without the limitations of NL-based methods. The work highlights that unsupervised EL abstractions can effectively enhance exploration in RL, bridging Embodied AI and Emergent Communication research.

**Strengths:**

The utilization of emergent language (EL) represents a relatively recent idea in the field.

**Weaknesses:**

1. Although this paper introduces the concept of Emergent Language (EL), many of its implementation details are not described in the main paper, and the overall exposition lacks organization, making several parts very difficult to follow. In particular, the most crucial component, the unsupervised referential games, is not clearly explained, nor are the motivations for using Eq. 4, or for employing descriptive distractors and data augmentation during referential game training. Overall, the paper does not successfully describe its core ideas, and the narrative is overly verbose and in need of substantial revision.

2. It remains unclear why data augmentation only uses Gaussian blur. Does blur offer a significant advantage over other perturbations such as noise? A more detailed justification is necessary.

3. The baseline methods used for comparison are outdated. For instance, RIDE and RND are approaches proposed around 2020 or earlier. The authors should demonstrate the effectiveness of their method by comparing it with more state-of-the-art techniques. This issue also reflects insufficient literature review in the related work section.

4. The authors claim that their method can be applied to various on-policy and off-policy algorithms, yet the experimental evaluation involves too few agent types. Moreover, it would strengthen the paper to include experiments with more recent models, such as Qwen or GPT-5.

5. The paper lacks an analysis of how the individual performance of the speaker and listener agents affects the final outcomes.

**Questions:**

1. Why are so many implementation details placed in the appendix instead of the main paper?

2. Why does the method not incorporate a wider variety of data augmentation techniques?

3. Why are more recent baseline methods not included for comparison?

4. How do the individual performances of the speaker and listener agents influence the final results?

---

### Official Review · Reviewer_GNgo · 2025-10-31

**Soundness:** 3
**Presentation:** 2
**Contribution:** 3
**Rating:** 6
**Confidence:** 3

**Summary:**

The paper tackles the exploration-exploitation challenge in Reinforcement Learning (RL) for Embodied AI agents by leveraging Emergent Language (EL) as state abstractions, addressing natural language’s (NL) limitations in availability and expressiveness. It proposes the EReLELA architecture, which employs unsupervised referential games to learn EL for intrinsic reward generation in count-based exploration. A Compactness Ambiguity Metric (CAM) is introduced to evaluate state abstractions. Experiments on MiniGrid environments show EReLELA achieves comparable sample efficiency to NL-based oracles and state-of-the-art methods, demonstrating EL as a viable alternative for hard exploration tasks.

**Strengths:**

**Language Abstractions Metric**: The proposed CAM metric quantitatively assesses state abstraction quality by analyzing temporal consistency in captions over video-like trajectories, filling a gap in existing literature (Sec. 3.2). Internal validation shows CAM effectively discriminates between languages with known abstraction differences and rejects inconsistent abstractions (e.g., shuffled captions), ensuring metric reliability. Application of CAM reveals that EL abstractions align more closely with task-relevant features compared to NL, highlighting their meaningfulness (Sec. 4.2).

**Effective Integration of RL Exploration**: EReLELA achieves competitive performance in hard-exploration environments like MultiRoom-N7-S4 and KeyCorridor-S3-R3, outperforming RND and matching RIDE in sample efficiency (Fig. 2). The agent successfully learns systematic exploration and object manipulation skills, reaching up to 80% success rates in KeyCorridor-S3-R2, validating the utility of EL abstractions (Fig. 3). By combining simple count-based exploration with EL abstractions, EReLELA reduces reliance on complex exploration algorithms, simplifying implementation (Sec. 1 and Sec. 3.1).

**Comprehensive Ablation Studies**: Ablations on RG loss functions (Impatient-Only vs. STGS-LazImpa) demonstrate how structural properties like Zipf's Law of Abbreviation impact RL performance (Sec. 4.2). Comparison between shared and agnostic encoder configurations reveals performance differences, highlighting potential gradient interference issues (Sec. 4.2).

**Weaknesses:**

**Experimental Scope and Generalizability**: Experiments are limited to 2D MiniGrid environments; results on 3D or real-world embodied environments (e.g., Habitat) are not presented, so generalizability is unclear (Sec. 4). Only two main environments are tested, and tasks with different exploration requirements (e.g., continuous action spaces) are not explored, limiting validation breadth (Sec. 4). While sample efficiency is shown, EReLELA’s final performance is only compared to RIDE, not to a wider range of modern exploration methods (e.g., Dreamer), so its standing among latest approaches is unclear (Fig. 2).

**Limited Analysis of Shared Encoder Interference**: Shared encoder versions underperform agnostic ones, but the paper does not empirically analyze gradient conflicts or representation overlaps (Sec. 4.2). No ablation is provided to isolate the effects of multi-task learning (e.g., gradient norms or loss weighting) between RL and RG objectives (Sec. 3.1). The discussion of interference remains speculative, without quantitative evidence from representation similarity measures (Sec. 4.2).

**Questions:**

1.	How does the CAM metric account for environments where temporal correlation does not directly correlate with abstraction quality? For instance, in dynamic environments with frequent state changes, could CAM misinterpret short caption intervals as poor abstraction?
2.	Given the underperformance of shared encoder configurations, have you considered alternative multi-task learning strategies (e.g., auxiliary task scheduling or gradient masking) to preserve RG and RL performance simultaneously?
3.	The paper uses a fixed number of distractors (K=256) during RG training. Is there a principled method for selecting K based on environment complexity, and how does K scale with larger observation spaces?

---

### Author Response · Authors · 2025-12-04
**Rebuttal (1/2)**

We thank all reviewers for their constructive feedback. Below we address each concern systematically, detailing the revisions made and acknowledging remaining limitations.

# 1. Limited Experimental Scope and Environment Diversity (Reviewers GNgo, wjMv, 5gbL)

Reviewer GNgo: "Experiments are limited to 2D MiniGrid environments; results on 3D or real-world embodied environments (e.g., Habitat) are not presented."

Reviewer 5gbL: "The algorithm has been evaluated on only two environments from MiniGrid... I suggest running the method on more environments from MiniGrid with varying levels of difficulty, and then on more complex environments such as Craftax-Classic or Craftax."

## Response:

We have expanded our evaluation to include the 3D MiniWorld FullMazeS5 environment, where EReLELA achieves new state-of-the-art performance (Figure 2). Our evaluation now spans four environments in the main text, and five in total with the appendices: MultiRoom-N7S4, KeyCorridor-S3R3, KeyCorridor-S6R3, and FullMazeS5 (and KeyCorridor-S3R2 in the appendices). The 3D results demonstrate robustness to high-dimensional visual observations. We acknowledge that evaluation on Habitat, Craftax, or continuous action spaces would further strengthen generalizability claims and flag this as important future work, but we consider it outside of the scope of our current analysis which focuses on showing feasibility of using EL for RL exploration and the critical component of it.

# 2. Outdated Baselines and Incomplete Comparisons (Reviewers GNgo, Jzmy)

Reviewer Jzmy: "The baseline methods used for comparison are outdated. For instance, RIDE and RND are approaches proposed around 2020 or earlier."

Reviewer GNgo: "EReLELA's final performance is only compared to RIDE, not to a wider range of modern exploration methods."

## Response:
We have substantially updated our baseline comparisons to include recent state-of-the-art methods:

1. ETD (Jiang et al., 2025) — the current best-performing method on our benchmarks
2. DEIR (Wan et al., 2023)
3. FirstVisit-Count adapted from Henaff et al. (2023) and Wang et al. (2023)

EReLELA outperforms or matches these recent methods across environments, achieving SOTA on FullMazeS5 (Section 4, RQ1).

# 3. Poor Presentation, Organization, and Clarity (Reviewers Jzmy, wjMv, 5gbL)

Reviewer wjMv: "The experimental section (particularly Section 4.1) is hard to follow. It mixes hypotheses, experiment descriptions, and results in a disorganized way."

Reviewer Jzmy: "The overall exposition lacks organization, making several parts very difficult to follow... the narrative is overly verbose and in need of substantial revision."

## Response:
We have restructured the experimental section around 3 clear research questions (RQ1, RQ2, RQ3), each with focused methodology and results. We have streamlined the narrative to emphasize our core contribution: that Relative Expressivity (RExpr) is the critical determinant of RL success (R² ≈ 0.52). We acknowledge that implementation details remain in the appendix due to space constraints; however, the main text now provides a self-contained description of the core method and findings, and we argue that it is difficult to tailor to every background levels when writing a paper that bridges two or more subfields, here being hard-exploration in RL and Emergent Communication.

# 4. Insufficient Analysis of Shared Encoder Underperformance (Reviewer GNgo)

Reviewer GNgo: "Shared encoder versions underperform agnostic ones, but the paper does not empirically analyze gradient conflicts or representation overlaps."

Reviewer GNgo: "The discussion of interference remains speculative, without quantitative evidence from representation similarity measures."

## Response:
All experiments in the main paper now show results of the agnostic case.
We acknowledge this limitation remains to be addressed possibly in future works or in the final camera-ready version (if acceptance). Our previous discussion of gradient interference between RL and RG objectives is indeed speculative, we have therefore moved that aspect to the appendices, in order to make the points of the main text less cluttered.
Quantitative analysis using gradient norms, representation similarity measures, or systematic loss weighting ablations would strengthen this aspect.
We have noted this as future work and believe it constitutes a valuable research direction, though it extends beyond the core contribution of this paper.

---

> ### Author Response · Authors · 2025-12-04
> **Rebuttal (2/2)**
>
> # 5. Missing Justifications for Design Choices (Reviewers Jzmy, wjMv)
>
> Reviewer Jzmy: "It remains unclear why data augmentation only uses Gaussian blur. Does blur offer a significant advantage over other perturbations such as noise?"
>
> Reviewer wjMv: "Unclear motivation for changing the RL algorithm. The paper switches between IMPALA and R2D2 without justification."
>
> ## Response:
> We have clarified the RL algorithm usage: all comparisons now use matched base algorithms (PPO or IMPALA) for fair comparison, with explicit labels in Figure 2. Regarding Gaussian blur, we selected it following Dessì et al. (2021) and found it sufficient for object-centric representations in practice. We acknowledge that a systematic comparison of augmentation strategies would be informative but consider it orthogonal to our main contribution regarding emergent language abstractions.
>
> # 6. Weak Link Between Hypotheses and Experimental Evidence (Reviewer wjMv)
>
> Reviewer wjMv: "The link between hypotheses and experiments is weak: H2 is supposed to concern the relationship between SNL and EL abstractions, but the paper merely states 'we therefore validate hypotheses (H1) and (H2)' without explaining how the results actually support H2."
>
> Reviewer wjMv: "Contradictory passages: for instance, the paragraph introducing the hypotheses states that H2 will use CAM, but in the evaluation paragraph it is H3 that uses CAM."
>
> ## Response:
> We have completely restructured the experimental section, replacing the confusing H1/H2/H3 framework with 3 clear research questions (RQ1–RQ3). Each RQ is now directly tied to specific experiments and results: RQ1 addresses performance comparisons, RQ2 analyzes abstraction types via CAM barycentric plots, and RQ3 examines the impact of RG training. This structure provides clear evidence-to-claim mappings.
>
> # 7. Missing Analysis of Speaker/Listener Agent Performance (Reviewer Jzmy)
>
> Reviewer Jzmy: "The paper lacks an analysis of how the individual performance of the speaker and listener agents affects the final outcomes."
>
> Reviewer Jzmy: "How do the individual performances of the speaker and listener agents influence the final results?"
>
> ## Response:
> We have introduced Relative Expressivity (RExpr) as a unifying metric that captures the quality of the speaker's abstractions and demonstrates strong correlation with RL success (R² = 0.52) and coverage (R² = 0.46). We show that controlling RExpr above 40% is critical for effective exploration (Section 3.1, RQ3). While this addresses the language-level impact, we acknowledge that decomposing individual speaker versus listener contributions remains unexplored and could provide additional insights into the communication dynamics.
>
>
> # Summary
>
> We believe the revised manuscript substantially addresses the core concerns regarding baselines (now including 2023–2025 methods), environment diversity (now including 3D), and presentation clarity (restructured around RQs). We reframed the main paper to only address the agnostic case in order to clarify our claims and remove the need for a deeper analysis of shared encoder interference, which has been pushed in the appendices. Individual RG agent contributions could be further addressed than through the RExpr concerns, and thus we acknowledge it as a remaining limitation. We hope to address it further in the extra page available if the paper is accepted.
> We thank the reviewers for their guidance in strengthening this work.

---

### Meta-Review · Area_Chair_sLRs · 2026-01-06

**Summary:**

1) Multiple reviewers expressed concern that the evaluation is confined almost entirely to a small set of 2D MiniGrid environments. While results are promising within this narrow setting, the lack of experiments on more diverse, complex, or realistic environments (e.g., 3D embodied settings, continuous action spaces, or real-world simulators) makes it difficult to assess whether the approach generalizes beyond toy domains.

2) Reviewers repeatedly noted that comparisons are primarily against older exploration methods (e.g., RND, RIDE). The absence of stronger or more recent baselines (e.g., Dreamer-style agents or other modern exploration approaches) weakens the claim that EReLELA is competitive with the state of the art.

3) Some reviewers also pointed out inconsistencies in the choice of RL backbone (e.g., switching between IMPALA and R2D2), making it harder to attribute gains specifically to the proposed method.

4) Several reviewers questioned whether the experimental evidence fully supports stronger claims, such as clear stage-wise reasoning decomposition, alignment between emergent language and task semantics, or conclusions about encoder interference. In particular, claims about shared-encoder gradient interference and layer-wise or stage-wise interpretations were seen as speculative without direct empirical validation

**Reviewer Concerns:**

1) During the rebuttal phase the authors have added experiments on a 3D MiniWorld environment (FullMazeS5) and expanded coverage to more MiniGrid variants. While inclusion of a 3D environment strengthens the empirical case and directly responds to reviewer requests, the evaluation is still limited to relatively small, synthetic environments; lack of results on more realistic simulators (e.g., Habitat, Craftax, continuous control) remains an open limitation, which the authors also acknowledge in the rebuttal. To quote: "We acknowledge that evaluation on Habitat, Craftax, or continuous action spaces would further strengthen generalizability claims and flag this as important future work, but we consider it outside of the scope of our current analysis which focuses on showing feasibility of using EL for RL exploration and the critical component of it." In my opinion, however, such an an experimental evaluation is not out-of-scope, which makes the experiments incomplete.

2) This has been resolved by the rebuttal and more recent baselines have been added, which are still outperformed. Specifically, when compared to the current state-of-the-art "Episodic novelty through temporal distance." Jiang et al. (ICLR 2025, although wrongly cited by the authors of the paper as an arXiv paper). However, the new comparison only uses the problem set proposed by the authors while omitting a full comparison on the benchmarks from Jiang et al.

3) The authors acknowledge that this witching was unintentional and corrected it. However, the lack of an original justification and analysis of backbone effects contributed to reviewers' confusion and weakens confidence in the experimental design. It seem like the experimental design was not completely fleshed-out at the time of submission.

4) The authors acknowledge this and say "We acknowledge this limitation remains to be addressed possibly in future works or in the final camera-ready version". However, these deferrals to future work are starting to pile up. Again, pointing to a manuscript that is not yet ready.


Overall, the method described in the paper seems to be rather promising. However, there are just too many things piling up that can be improved for the paper to be accepted at ICLR.

**Reviewer Scores:**

It is difficult to say none of the reviewers engaged in the rebuttal process. Nothing.

---

### Decision · Program_Chairs · 2026-01-26

Reject